# Carrageenan: Drug Delivery Systems and Other Biomedical Applications

**DOI:** 10.3390/md18110583

**Published:** 2020-11-23

**Authors:** Edisson-Mauricio Pacheco-Quito, Roberto Ruiz-Caro, María-Dolores Veiga

**Affiliations:** Department of Pharmaceutics and Food Technology, Faculty of Pharmacy, Complutense University of Madrid, 28040 Madrid, Spain; edissonp@ucm.es (E.-M.P.-Q.); mdveiga@ucm.es (M.-D.V.)

**Keywords:** carrageenan, marine polysaccharide, biological activity, drug delivery

## Abstract

Marine resources are today a renewable source of various compounds, such as polysaccharides, that are used in the pharmaceutical, medical, cosmetic, and food fields. In recent years, considerable attention has been focused on carrageenan-based biomaterials due to their multifunctional qualities, including biodegradability, biocompatibility, and non-toxicity, in addition to bioactive attributes, such as their antiviral, antibacterial, antihyperlipidemic, anticoagulant, antioxidant, antitumor, and immunomodulating properties. They have been applied in pharmaceutical formulations as both their bioactive and physicochemical properties make them suitable biomaterials for drug delivery, and recently for the development of tissue engineering. This article provides a review of recent research on the various types of carrageenan-based biomedical and pharmaceutical applications.

## 1. Introduction

The oceans make up the majority of the Earth’s surface and host a wide range of marine organisms that supply a rich deposit of very valuable material resources, including algae, crustaceans, and other microorganisms that provide compounds called marine biopolymers, which are distributed in three important groups: polysaccharides, proteins, and lipids [1].

Biopolymers are polymers produced from natural sources and can be biosynthesized from living organisms, such as algae, or chemically synthesized from biological material [2]. Due to their chemical composition, physical behavior, wide variety, and relatively low cost, these renewable resources have aroused great interest in the pharmaceutical, biomedical, cosmetic, and food industry [3,4]. Biopolymers have been used for decades in the food and cosmetic industry, and in recent years have been studied in pharmaceutical applications as functional excipients, bioactive ingredients, and in tissue engineering [3,5,6,7,8]. 

Marine polysaccharides are the priority focus in the study of biopolymers due to their crucial importance in view of their sources and their easy acquisition as a renewable resource. The marine plant polysaccharides, specifically the seaweed polysaccharides, are considered the most abundant polysaccharides in marine organisms, and they are widely distributed in the ocean, from tidal level to considerable depths, free-floating or anchored. The content of polysaccharide present in seaweed is high, accounting for more than 50% of the dry weight [9,10]. The main components of the cell walls of red seaweed are carrageenans (CGs) and represent between 30% and 75% of the algal dry weight [11]. These polysaccharides are considered the most widely studied red algae polysaccharides [12].

Carrageenans are linear sulphated polysaccharides that are extracted from various genera of red algae in the Florideophyceae class: *Agardhiella*, *Chondrus crispus*, *Eucheuma*, *Furcellaria*, *Gigartina*, *Hypnea*, *Iridaea*, *Sarconema*, and *Solieria* [13,14,15]. 

The use of CGs was first described in Ireland in the early 19th century. The word “carrageen” was introduced around 1829, and probably came from Carrigan Head in Co. Donegal in north-western Ireland. “Carrigan or carrageen” is a common name throughout Ireland meaning “little rock” [16]. The first types of CG were obtained from *Chondrus crispus*, which was used for animal feed and medicinal purposes. There is also archaeological evidence showing that the Chinese used carrageenan-bearing macroalgae, such as *Gigartina* spp., for similar purposes in around 600 B.C. [16,17]. 

CG were initially used as a thickening agent in the food industry, and due to their gelling, emulsifying, and stabilizing properties have been utilized until the present day in a wide range of fields [13,14,18]. CGs are “Generally Recognized As Safe” (GRAS) by the Food and Drug Administration (FDA) since 1973 (FDA SCOGS (Select Committee on GRAS Substances). Carrageenan (E-407) and semi-refined carrageenan (E407a) have been approved by the European Food Safety Authority as food additives. The toxicological aspects of CGs have been thoroughly evaluated and they have been established to have minimal or no adverse physiological effects [19]. 

Due to their biocompatibility, high molecular weight, high viscosity, and gelling capacity, these polymers have gained great importance in recent decades not only in the food industry but also in medical, pharmaceutical, and biotechnological research. Figure 1 shows the number of publications (reviews and research articles) retrieved in the ISI Web of Knowledge with the keyword “carrageenan” from 1970 to the present, evidencing the growing interest in CGs since the 1990s, when the volume of scientific publications on these polysaccharides began to grow. In the last two years, the trend has been maintained, with 776 scientific publications in 2018 and 786 scientific publications in 2019, and 265 scientific publications until April 2020.

In Figure 2, the total number of publications was filtered by category, with the highest percentages in pharmacology and pharmacy (27.00%), food science technology (16.19%), medicinal chemistry (12.69%), applied chemistry (12.06%), and polymer science (8.97%), demonstrating the importance of carrageenan in pharmaceutical and biomedical applications (Figure 2).

CG has therefore become one of the main biomaterials for a range of purposes in the pharmaceutical industry, since they have been used to improve the formulation of medications and the controlled release of drugs [20,21,22]. CG has also revealed a range of bioactive applications that have been studied, including their role as antioxidant [23,24,25], antiviral [26,27,28,29], antibacterial [30], antihyperlipidemic [31], anticoagulant [25,26], antitumor, and immunomodulatory agents [32,33,34], enabling them to be used in the biomedical field as potential pharmaceutical formulations for the treatment of various diseases.

Consequently, in view of the various applications of CG, the aim of this review is to present a general description of the recent and most relevant aspects relating mainly to their biological activities and pharmaceutical applications.

## 2. General Properties of Carrageenan

### 2.1. Chemical Structure and Properties 

CG is the general name of a group of high-molecular-weight hydrophilic sulphated polysaccharides formed by alternate units of d-galactose and 3,6-anhydro-galactose (3,6-AG) joined by alternating α-1,3 and β-1,4-glycosidic bonds. It contains between 15% and 40% of ester-sulphate, meaning it is an anionic polysaccharide. CG can be classified into six basic forms depending on their sulphate content, source of extraction, and solubility: *kappa* (κ-), *iota* (ɩ-), *lambda* (λ-), *mu* (μ-), *nu* (ν-), *beta* (β-), and *Theta* (θ-) carrageenan. The most important types of CG applied in the pharmaceutical and commercial field are *kappa*-carrageenan (*kappa*-CG), *iota*-carrageenan (*iota*-CG), and *lambda*-carrageenan (*lambda*-CG). These are generally obtained separately or as a well-defined mixture, since most seaweeds contain hybrid CG [4,13,15]. 

*Kappa*-, *iota*-, and *lambda*-CG are distinguished by the presence of one, two, and three ester-sulphate groups per repeating disaccharide unit, respectively. The chemical structures of CG are heterogeneous and are correlated to the algae sources, the life stage of the seaweed, and the extraction processes [9]. Their structures and other characteristics are shown in Table 1. 

The solubility of CG is influenced by various factors, such as temperature, pH, the presence of other solutes, the type of carrageenan (sulphate groups), and their associated cations (K^+^, Ca^++^). CGs demonstrate solubility in hot water. Sodium salts of *kappa*-CG and *iota*-CG are also soluble in cold water, although potassium salts are not. CGs are also insoluble in organic solvents, such as alcohol, ether, and oil [4,36,37]. 

Commercial CGs are generally presented as sodium, potassium, and calcium salts or a mixture of these. These salts provide stability, and the average molecular mass of commercial CG varies between 100 and 1000 kDa. Sodium forms of carrageenan are more readily soluble, while potassium forms are less so [4,13]. 

Gel formation is one of the most important properties for the pharmaceutical and commercial application of CG. It has been observed that *kappa*-CG and *iota*-CG form three-dimensional gels through interactions with metal ions (potassium or calcium) that may be either clear or turbid, rigid or elastic, tough or soft, and heat stable or thermally reversible [13,20,38]. 

Potassium ions are the best gelling agent for *kappa*-CG, forming a rigid and brittle gel, while calcium ions are less effective for gelation. The combination of both ions, however, produces a strong gel. *Kappa*-CG gels are the strongest and most rigid, and synerise, while *iota*-CG gels best with calcium ions to produce soft and flexible gels with good freeze-thaw stability that do not synerise [38,39].

Gelation of CG occurs on cooling the solutions of *kappa*-CG and *iota*-CG; the linear molecules originate double or triple helices with a restricted length due to the absence of an α-d-galactopyranosyl unit containing a 3,6-anhydro ring. The linear helicoidal proportions therefore bind and form a firm and stable three-dimensional gel network with the appropriate ions (Figure 3) [38,40].

In contrast, rather than forming a three-dimensional gel, *lambda*-CG improves the viscosity of the medium and is used as a thickener. However, recent research has shown that it was possible to gel *lambda*-CG but with trivalent ions. This novel finding has the potential to extend the current utility of *lambda*-CG beyond a viscosifying agent [41].

### 2.2. Source and Production of Carrageenan 

CGs were originally extracted from some types of seaweed of the phylum Rhodophyta, until growing demand and the expansion of their use led to the cultivation of new species to ensure their year-round availability.

Most CGs are therefore extracted from *Kappaphycus alvarezii* and *Eucheuma denticulatum*. *Kappa*-CG is mostly extracted from *Kappaphycus alvarezii*, commercially known as *Eucheuma cottonii*, while *iota*-CG is predominantly produced from *Eucheuma denticulatum*, also known as *Eucheuma spinosum*. The advantage of these two species over the others that are traditionally used is the types of carrageenan that are extracted, since species, such as *Chondrus crispus*, contain a mixture of *kappa*- and *lambda*-CG that cannot be separated during commercial extraction. *Lambda*-CG is obtained from seaweed in the *Gigartina* and *Chondrus* genera [14,42,43].

The manufacturing of CG consists of extraction, purification, concentration, precipitation, filtration, and drying, although the process may vary according to the family of red algae used to extract the sulphated polysaccharide. Specific extraction methods are considered trade secrets by their manufacturers but generally follow a similar process [21,40]. There are several methods to extract the CG; we give a brief description of the most common. The original method for manufacturing CG is to extract the polysaccharides by means of aqueous solutions, filtering to eliminate the remaining residues, recovering the solution by precipitation using alcohol, and finally separating, drying, and milling the precipitate to produce refined carrageenan. This method is time-consuming and energy-intensive and does not have a high extraction efficiency, which is why more precise methods have been developed. Another method described in the literature is to extract the CGs as insoluble residue after washing out any residual minerals, soluble proteins, and lipids. This insoluble residue is sold as semi-refined low-purity carrageenan [40,44,45].

Other more complex methods include extraction using enzyme [46,47], or fungal [48], extraction with deep eutectic solvents (DESs) [49,50]. Microwave-assisted extraction, ultrasound-assisted extraction, reactive extrusion, and photobleaching processes have also been reported in recent years [51,52,53,54]. Each extraction method may offer certain advantages over traditional methods, enabling the improvement of the physico-chemical, gelling, and bioactive conditions of the CG. Finally, the variables used in the extraction process, such as temperature, pH, time, and alkaline pre-treatment, must always be taken into account, since these will condition the expected results [55].

## 3. Bioactive Properties of Carrageenan

CGs have shown potential bioactive qualities, including antiviral, antibacterial, antihyperlipidemic, anticoagulant, antioxidant, antitumor, and immunomodulatory properties [5,13,18]. Figure 4 shows a summary of these properties and the carrageenan types that display them.

### 3.1. Antiviral Activity

The antiviral activity of sulphated polysaccharides extracted from red algae was first demonstrated by Gerber et al. when they observed that polysaccharides obtained from *Gelidium cartilagenium* protected embryonic eggs against the influenza B or mumps virus [56]. 

The chemical structure, degree of sulphation, distribution of the sulphate groups, molecular weight, constituent sugars, conformation, and dynamic stereochemistry determine the antiviral activity of the sulphated polysaccharides [5]. The antiviral effect of CG is due to the specific screening of the cellular structures involved in binding the virus to its receptor [57]. Thus, for example, the antiviral activity of *lambda*-CG may be due to the irreversible formation of stable virion-CG complexes, and therefore, the viral envelope sites essential for the binding of the virus to the host cells are occupied by carrageenan, preventing the virus from completing the infectious process (Figure 5) [58,59]. 

CG is a selective inhibitor of several viruses, including the herpes simplex virus (HSV), the human papillomavirus (HPV), the varicella zoster virus (VZV), human rhinoviruses, and others [4,5,20,45,59,60]. 

Several in vitro/in vivo studies have reported that CGs act as potent and selective inhibitors of the herpes simplex virus-1 (HSV-1) and the herpes simplex virus-2 (HSV-2) [27,54,61]. The combination of CG with lectin has also shown strong activity against HSV-2 and HPV in in vitro and in vivo studies [62,63]. CG has demonstrated activity against HPV, and is 1000 times more effective against HPV in cell culture tests than against HSV and HIV. Carrageenan acts primarily by preventing the binding of HPV virions to cells [28]. 

CGs have been applied with other active ingredients to enhance or direct their bioactivity against HPV and HSV-2. An intravaginal ring was reported containing four active ingredients: *lambda*-/*kappa*-CG, zinc acetate, levonorgestrel, and MIV-150 (microbicide); this formulation had a prolonged action and allowed protection against HIV-1, HSV-2, HPV, and unwanted pregnancies [64]. 

More recently, Perino et al. evaluated the safety, satisfaction, and antiviral effect of a new carrageenan-based vaginal microbicide in a population of fertile patients with genital human papillomavirus. The gel was formulated with 0.02% from different types of CG and *Propionibacterium* extract, and it was administered in two treatment phases. During the first phase, gel therapy was applied once daily for 30 days continuously; the second phase began with an application on alternate days for 45 applications. In both phases, the patients could also have sexual intercourse without a condom. The results showed that 60% of the patients presented negative HPV; the gel was safe and well tolerated by women. This research also supports the hypothesis that CGs play a role in accelerating the normal clearance of genital HPV infection in women with a positive HPV-DNA test [65].

*Iota*-CG was active against respiratory viruses in vitro and was effective as a nasal spray in clinical trials. Eccles et al. investigated *iota*-CG in patients with early common cold symptoms. Their results indicated a significant reduction in cold symptoms in the *iota*-CG group compared to placebo during the first four days when symptoms were most severe [66], and also corroborated that *iota*-CG reduces the growth of human rhinoviruses (HRVs) and inhibits the virus-induced cytopathic effect of infected HeLa cells. Consequently, it is observed that *iota*-CG, like the other types of CG, acts by preventing the binding or entry of virions into host cells [29,66]. 

*Iota*-CG was tested as a potential inhibitor of the influenza A virus infection. The inhibitory potential of *iota*-CG with IC50 values of around 0.2 µg/mL in H1N1 and 0.04 µg/mL in H3N2 infections was up to 10 times higher than with *kappa*-CG. The authors thus confirmed that *iota*-CG reduced the spread of the influenza virus in the surface epithelia of infected animals, providing sufficient benefit for animals to promote survival. Clearly, this research determined that *iota*-CG was safe and effective in the treatment of influenza infection, even above other types of CG, making it a suitable antiviral candidate for the prophylaxis and treatment of this infection [67]. 

Shao and co-workers investigated the ability of *kappa*-CG to inhibit the swine flu pandemic 2009 H1N1 influenza virus. The results demonstrated that *kappa*-CG could significantly inhibit A/Swine/Shandong/731/2009 H1N1 (SW731) replication by interfering with a few replication steps in the SW731 life cycles, including adsorption, transcription, and viral protein expression. *Kappa*-CG inhibited SW731 mRNA and protein expression after internalization into cells. This indicates that *kappa*-CG may be suitable for use against H1N1/2009 and other similar viruses [68], but this is not always the case, since in the study previously described, *kappa*-CG was less effective against influenza A virus infection. This disadvantage related to antiviral efficacy could be related to various factors, such as the molecular weights, sulfate content, and the origin of carrageenans.

Many authors have proposed improving the antiviral effectiveness of CG by combining it with other drugs. The combination of an anti-influenza drug Zanamivir with CG in a nasal formulation was evaluated in vitro and in vivo, and the results indicated that CG and Zanamivir act synergistically against several influenza A virus strains (pandemic H1N1/09, H3N2, H5N1, H7N7). In this study, it can be observed that the combination of two types of carrageenan (*iota*-CG and *kappa*-CG) and an antiviral drug increase the efficacy of the formulation, since it was observed that the combined use of the compounds significantly increases the survival of infected animals in comparison with mono-therapies or placebo [69].

A recent study reported the evaluation of a nasal spray containing xylometazoline HCl and *iota*-CG. In vitro experiments revealed that the combination of the vasoconstrictive properties of the drug and the antiviral activity of *iota*-CG were effective against human rhinovirus (hRV) 1a, hRV8, and human coronavirus OC43. In this study, it was clearly observed that the combination of *iota*-CG with the drug does not produce a negative effect, but rather the efficacy and safety of both components remain unchanged, producing the desired therapeutic effect, showing that the combination of CG with drugs fails to produce harmful interactions. The formulation was well tolerated at the application site, with no occurrence of erythema or edema in the nostrils of any of the rabbits or any signs of toxicity in any of the organs and tissues examined [70]. These studies generally mentioned that CG created a physical barrier in the nasal cavity against respiratory viruses, preventing their binding to host cells [57]. 

Chiu et al. reported that *kappa*-CG has a strong and effective anti-enterovirus 71 (EV 71) activity able to reduce plaque formation, prevent viral replication before or during viral adsorption, and inhibit EV 71-induced apoptosis. In the virus binding assay, they demonstrated that *kappa*-CG can bind firmly to the EV 71 to form CG-virus complexes, so the virus–receptor interaction (Figure 5) is likely to be disrupted [71]. 

The potential role of *lambda*-CG P32 on the inhibition of the rabies virus (RABV) has also been studied, and the results show it to be a promising anti-RABV agent that can effectively inhibit RABV infection in vitro by affecting viral internalization and cell fusion mediated by viral G protein. A comparison between *lambda*-CG P32 and the P32 structural analogues (heparan sulphate and heparin) revealed that the effect of P32 inhibition on RABV infection was stronger than heparan sulphate and heparin, suggesting that the notable anti-RABV activity of P32 can be attributed to more than its structural similarity to heparan sulphate. The authors also investigated the inhibitory effect of *lambda*-CG on the vesicular stomatitis virus (VSV), and observed a null inhibition [72].

A more recent study by Abu-Galiyun et al. reported the inhibitory effects of *kappa*-CG, *iota*-CG, *lambda*-CG, and other natural polysaccharides on VZV infection in vitro. Almost all the polysaccharides tested were very active against VZV compared to acyclovir as a reference drug and exhibited dose-dependent behavior. The results suggested that *iota*-CG may inhibit the early step/s of the virus infection, such as virus attachment or penetration to the host cells, and the late step/s after the penetration of the virus into the host cells, showing that *iota*-CG has strong antiviral activity on various types of viruses [73]. 

Finally, a recent study, developed during the pandemic caused by the severe acute respiratory syndrome coronavirus 2 (SARS-CoV-2), determined that marine sulfated polysaccharides, such as *iota*-CG, can inhibit viral binding and entry into host cells. *Iota*-CG could prevent the infection at concentrations ≥125 μg/mL [74]. Therefore, we observe that marine polysaccharides have adequate antiviral activity in various viruses, inhibiting viral binding, which allows their use in the treatment and prevention of Coronavirus disease 2019 (COVID-19) [74,75,76]. 

In general, studies have shown that the antiviral efficacy of CG is very broad, which would make it possible to suppress the replication of various viruses with or without envelopes. The antiviral effect of CG occurs alone or in combination with other compounds, such as drugs, and this combination allows substantial improvement of the therapeutic efficacy and generally does not produce interactions. The antiviral activity of CG continues to be promising since various viral strains, such as SARS-CoV-2, continue to be analyzed, and these polysaccharides fulfill this proposed antiviral efficacy, but there is still a long way to go in order to fully evaluate the efficacy and safety of these polysaccharides as highly effective antiviral components.

### 3.2. Antibacterial Effects

CG can inhibit infection caused by a variety of bacteria. Yamashita et al. evaluated the antimicrobial action of dietary polysaccharides on foodborne pathogenic bacteria and showed that CG had the most pronounced inhibitory effect of all polysaccharides, significantly inhibiting the growth of almost all the bacterial strains studied. A growth-inhibition experiment using *Salmonella enteritidis* showed that the inhibitory effect of the CG was bacteriostatic. The investigation also indicated that the removal of sulphate residues eliminates the bacteriostatic effect of *iota*-CG, suggesting that the sulphate residue (s) in CG play an essential role in this effect [77], being very similar to that observed in the antiviral effect; thus, the sulfate content and the molecular weight are characteristics that determine the bioactive capacity of CG.

In other research, Inic-Kanada and co-workers tested the effects of *iota*-CG on ocular *Chlamydia trachomatis* infection. *Iota*-CG tends to reduce the infectivity of *Chlamydia trachomatis* in vitro and when it was evaluated in vivo, the results showed slightly reduced ocular pathology and significantly less shedding of infectious elementary bodies, which suggested that *iota*-CG could be a promising agent for reduction of the transmission of ocular chlamydial infection, with more in-depth studies to support these first results [30]. 

Another way to evaluate the antibacterial efficacy of carrageenans was through the use of *kappa*-CG oligosaccharides against *Escherichia coli*, *Staphylococcus aureus*, *Saccharomyces cerevisiae*, *Penicillium citrinum*, and *Mucor* spp., determining the antibacterial efficacy by measuring the diameter of the inhibitory zone. Consequently, it was observed that all *kappa*-CG oligosaccharides had inhibitory activity against the bacteria studied, presenting a greater inhibitory activity against *Saccharomyces cerevisiae* [78].

Another work evaluated a modification of CG as an antibacterial agent (oxidized *kappa*-CG). The results mentioned that the oxidized *kappa*-CG could damage the bacterial cell wall and cytoplasmic membrane and suppress the growth of both Gram-positive and Gram-negative bacteria. Oxidized *kappa*-CG possessed broad-spectrum antibacterial activity and could be a suitable candidate for the development of a new antibacterial agent, requiring further studies [79].

*Kappa*-CG was added to a sinus rinse that is commercially available on the Australian market (Flo CRS^®^ and Flo Sinus Care^®^), and applied directly to cultured human primary nasal epithelial cells at the air–liquid interface, and to human bronchial epithelial cells in the presence of different *S. aureus* strains. The addition of *kappa*-CG to commercially available sinus rinses reduced the production of interleukin-6 in cells treated with *kappa*-CG and Flo Sinus Care^®^. The addition of *kappa*-CG to both Flo CRS^®^ and Flo Sinus Care^®^ rinses also reduced the intracellular infection rate by an average of 2% [80]. These results suggest that GC can be used in inhalation drug delivery systems, inhibiting infection caused by a facultative Gram-positive anaerobic bacterium. 

PVA, a hydrogel based on carrageenan crosslinked with silane, showed strong antibacterial activity against *S. aureus* and slight activity against *E. coli*. The antibacterial activity may be due to the interaction of CG molecules with the cell membrane of the bacterial strains, and more specifically with the structure of the CG, as carrageenan comprises negatively charged SO_4_^2−^ suspended groups, and *S. aureus* (Gram-positive) has an outer covering of mucopeptide and peptidoglycan lipids; whereas *E. coli* consists of phospholipids and lipopolysaccharides, which gives its surface a strongly negative charge. These interactive sites on the Gram-positive bacteria favor the alteration of the bacterial cell membrane that controls bacterial growth. The authors also indicate a second factor that could control bacterial growth, namely the bonding of CG and PVA with the DNA of the bacterial strain, thereby limiting transcription and translation by DNA [81].

A recent study described the carboxymethylation of *kappa*-CG with monochloroacetic acid to achieve different degrees of substitution of carboxymethyl-*kappa*-CG, in order to improve the properties of the polysaccharide. In antibacterial assays, carboxymethyl-*kappa*-CG with degrees of substitution of 0.8, 1.0, and 1.2 exhibited growth inhibition against *S. aureus*, *Bacillus cereus*, *E. coli*, and *Pseudomonas aeruginosa*. The antibacterial activity could be due to the presence of sulphate, and carboxylate groups may create an acidic pH environment; it is also possible that the carboxylate groups could increase the nucleophilicity of the polymer. Although the authors recommended further studies on this subject, they noted that antioxidant, antibacterial, and biocompatibility tests could confirm potential applications of this polymer, such as for wound dressings and scaffolds [82]. 

Studies that describe antibacterial effects tend to be scarcer compared to antiviral effects, and perhaps this is because in most cases, the antibacterial effect of carrageenan occurs when they are modified by various processes, such as oxidation or carboxymethylation, thus allowing fulfilment of the antimicrobial effect in few bacteria. 

### 3.3. Antihyperlipidemic Effects

CG also presents biological activity in the gastrointestinal tract, generally associated with oral administration. When ingested it increases the viscosity of the intestinal content and decreases the rate of digestion and absorption, which in turn reduces the diffusion of enzymes, substrates, and nutrients in the intestinal absorption phase, resulting in a lower absorption of nutrients, including cholesterol (the hypocholesterolemic effect) [45]. In other words, the bioactive potential of CG derives from its ability to decrease the cholesterol absorption rate and increase the rate of synthesis of endogenous cholesterol [83]. 

Several research works have demonstrated the bioactive role of CG, mainly *kappa*-CG, in reducing serum levels of total cholesterol, triglycerides, and low-density lipoprotein cholesterol (LDL-C), and increasing high-density lipoprotein cholesterol (HDL-C) in the peripheral blood, due to the interaction of their chemical structure in digestion [84,85].

Sokolova et al. reported the effect of *kappa*-CG, *kappa*/β-CG and *iota*/*kappa*-CG individually and in combination with lipopolysaccharide on the synthesis of prostaglandin E2 and cytokines (interleukin [IL]-1β and IL-6) in a whole blood model in vitro. At high concentrations, CGs have a substantial ability to modulate prostaglandin E2 synthesis and stimulate IL-1β and IL-6 synthesis, confirming the possible mechanism of the cholesterol-reducing properties of carrageenan [86].

In a recent study, the authors evaluated the bioactive potential of carrageenan in the lipid profile in individuals with total cholesterol levels equal to or higher than 200 mg/dL after the ingestion of a jelly composed of a hybrid *kappa*-CG/*iota*-CG polysaccharide. In total, 100 mL of jelly per day were ingested for 30 and 60 days. The results showed a statistically significant decrease in total cholesterol and HDL-C in both periods (30 and/or 60 days). Daily intake for 60 days also showed a reduction in serum levels of total cholesterol and LDL-C in women [87].

*Kappaphycus alvarezii* was used as a whole-food supplement to attenuate the development of obesity in rats fed a high-carbohydrate high-fat diet that mimics symptoms of human metabolic syndrome, including central obesity, hypertension, dyslipidemia, and impaired glucose tolerance, coupled with the cardiovascular and liver complications of metabolic syndrome. The study highlighted the potential of *Kappaphycus alvarezii* as a functional food with possible application for the prevention of metabolic syndrome. The researchers also demonstrated that *Kappaphycus alvarezii* may reverse metabolic syndrome through the selective inhibition of obesogenic gut bacteria and the promotion of health-promoting gut bacteria [88].

A new in vivo study investigated the potential of red seaweed (*Sarconema filiforme*) as a functional food for the reversal of metabolic syndrome and its possible mechanisms in male Wistar rats. Rats fed a high-carbohydrate high-fat diet supplemented with *S. filiforme* as a source of *iota*-CG decreased their body weight, systolic blood pressure, abdominal and liver fat, and plasma total cholesterol concentrations compared to controls. *Iota*-CG attenuates symptoms of diet-induced metabolic syndrome in rats. The correlations between changes in the gut microbiota and physiological changes following administration of *S. filiforme* suggest that the likely mechanism is that CGs act as prebiotics, and through systemic anti-inflammatory responses in organs, such as the heart and liver [89].

The antihyperlipidemic effect of CG is little studied and perhaps is due to the fact that the components that fulfill this effect, in general, must be included as food supplements in the diet, and in Western countries, these components are not used to a great extent, which leads to a waste of these properties. It should be mentioned that the main advantages at the metabolic level produced by CG are due to the fact that they are not degraded or absorbed in the gastrointestinal tract, allowing a decrease in the rate of digestion and absorption of all nutrients included in the diet. 

### 3.4. Anticoagulant and Antithrombotic Activity 

Anticoagulant activity could be influenced by the glycosidic bond (either (1 → 3) or (1 → 4)) and the neighboring sulphate groups [90]. Liang et al. studied the anticoagulant potential of different types of carrageenan and their derivatives: *kappa*-CG oligosaccharide, sulphated *kappa*-CG, and desulphated *kappa*-CG. *Lambda*-CG demonstrates the highest anticoagulant activity in the rabbit whole blood test, while *kappa*-CG oligosaccharide and desulphated *kappa*-CG show no anticoagulant activities. In the case of oligosaccharide, this is probably due to the lack of a secondary structure caused by the decrease in molecular weight, and in desulphated *kappa*-CG to the absence of favorable sulphate groups. The substitution position of the sulphate groups has a greater impact than the degree of substitution on both anticoagulant activity and cell proliferation. C-2 of 3,6-anhydro-α-d-Galp is the most favorable position for substitution, whereas C-6 of β-d-Galp is the most disadvantageous. Secondary structures of glycans also play a key role in biological activities [91]. 

Among carrageenan types, *lambda*-CG has approximately twice the activity of unfractionated carrageenan and four times the activity of *kappa*-CG. *Lambda*-CG showed greater antithrombotic activity than *kappa*-CG, probably because anticoagulant or antithrombotic activity is influenced by the number of sulphate groups present in the chemical structure. The main basis for carrageenan’s anticoagulant activity appears to be an antithrombotic property [4,21]. 

In a recent study, selective oxidation was performed on five CG-*kappa*-CG, *iota*-CG, *iota*/nu-CG, *Theta*-CG, and *lambda*-CG at C-6 of the β-d-Galp in order to evaluate the in vitro anticoagulant activity of oxidized derivatives. The results showed a synergic effect of the carboxyl groups on the anticoagulant activity, which was dependent on the regiochemistry of the sulphate groups in the polysaccharide backbone. Sulphate groups at C-2 of the β-d-Galp units appeared to positively influence the anticoagulant effect compared to C4-sulphate samples. The partially oxidized *kappa*-CG derivative also exhibited a better anticoagulant effect than the fully oxidized carrageenan. These results support the synthesis of new CG derivatives to increase anticoagulant properties [92]. These results show that CGs have anticoagulant and antithrombotic properties but require structural modifications that allow them to develop this bioactive activity. This leads to the development of chemical processes that perhaps hinder studies in this area, with few published works being observed.

### 3.5. Antitumor and Immunomodulatory Activity

Several studies have shown that CG has immunomodulatory and antitumor activity [93,94,95]. The antitumor activity of CG could be related to the destabilization of the interaction of the glycosaminoglycans (GAGs) portion of the proteoglycans and the extracellular matrix proteins, thus eliminating the adhesion of cancer cells to matrices, which is necessary for the spread of metastasis [96]. The bioactive properties of CG depend on their chemical structure, molecular weight, and the quantity and position of sulfation, so *lambda*-CG can be degraded into several products with different molecular weights, all exhibiting anticancer effects, probably through immunomodulation. The lower molecular weight compounds, 15 and 9.3 kDa, showed higher anticancer and immunomodulation effects [95,97].

CG has been considered as an adjuvant in cancer immunotherapy. Tumor-inhibiting activity of low-molecular-weight *lambda*-CG and its mixture with 5-Fluorouracil (5-Fu) on mice transplanted with S180 tumor was investigated. The results suggested that the degraded *lambda*-CG could enhance the antitumor activity of 5-Fu and improve immunocompetence damaged by 5-Fu [93]. *Lambda*-CG was also reported to inhibit tumor growth in mice with murine melanoma cell lines and murine mammary tumor cell lines through intratumoral injection. *Lambda*-CG exhibited an efficient adjuvant effect in an ovalbumin-based preventative and therapeutic vaccine for cancer treatment, which significantly enhanced the production of anti-ovalbumin antibody. The toxicity analysis suggested that *lambda*-CG had a good safety profile. Therefore, this study suggests that *lambda*-CG could be used as a potent antitumor agent and as an adjuvant in cancer immunotherapy [98].

The effects of CG on the tumor cell cycle were investigated using human cervical carcinoma cells (HeLa) and human umbilical vein endothelial cells (HUVECs). The results indicated that CG interrupted the cell cycle at specific stages and delayed the time necessary for the cell to progress through the cell cycle. *Kappa*-CG was found to delay the cell cycle in the G2/M phase, while *lambda*-CG stalled the cell cycle in both the G1 and G2/M phase. *Lambda*-CG also suppressed the cell’s ability to divide, demonstrating a strong antiproliferative effect [99]. Degraded *iota*-CG suppressed tumor growth, induced apoptosis, and halted the G1 phase, which improved the survival rate of tumor-bearing mice [100].

Calvo et al. tested purified native and degraded CGs and their disaccharides, obtained from extracts of the potential cytotoxic and antitumor compounds *Hypnea musciformis*, *Iridaea undulosa* and *Euchema spinosumas*. The results showed that *kappa*-CG and *iota*-CG and carrageenan oligosaccharides had a cytotoxic effect on LM2 tumor cells. Some oligosaccharides are also more cytotoxic than their parent compounds, indicating that a lower molecular weight is one of the factors that improves its cell ability. These results point to the potential use of disaccharide units, such as carrabioses coupled to antineoplasics, to improve their cytotoxicity and antimetastatic properties, and the use of *iota*-CG as an adjuvant or carrier in anticancer treatments [101]. 

In a recent study, Cotas et al. showed that the CGs extracted from the two *Gigartina pistillata* life cycle phases, particularly the T (tetrasporophyte) carrageenan, have potential against colorectal cancer stem-like cells. This could be explained by the higher content of sulphated ethers in T carrageenan (*lambda*-CG/*epsilon*-CG) compared to the female gametophyte carrageenan (*kappa*-CG/*iota*-CG) [102].

Antitumor and immunotropic effects were reported in *kappa*-CG and *lambda*-CG isolated from *Chondrus armatus* and their low-molecular-weight degradation products. The results showed that low-molecular-weight CG degradation products not only retain the biological activity of their high-molecular-weight precursors but also increase their efficacy in a type-dependent manner. CG degradation is a viable solution to increase their biomedical applicability by overcoming the limitations of their chemical and physical properties [103].

Within the antitumor activity, it is completely clear that the effect is fulfilled when CGs are modified or degraded, since these processes have been a viable solution to increase the biomedical applicability of CGs, overcoming certain limitations of their chemical and physical properties. Furthermore, many studies agree that low molecular weight influences the therapeutic efficacy of CG.

### 3.6. Antioxidant Activity 

Studies carried out in recent years have shown that CG also has significant antioxidant activity, a property associated with sulphate group content [82,104]. Gomez-Ordoñez et al. tested CG sequentially extracted from *Mastocarpus stellatus* with water, acid, and alkali. Extraction with water produced polysaccharides with the highest degree of sulfation and the highest molecular weight. The analysis of these polysaccharides showed that they had the best results in terms of in vitro antioxidant and anticoagulant capacity, confirming that the number of sulphate groups influences antioxidant activity, and that a high molecular weight plays a role in anticoagulant capacity. These authors found that the different extraction methods influence the bioactive capacity of the CG [25]. Another study demonstrated the antioxidant capacity of a multilayer coating based on *kappa*-CG and lecithin/chitosan loaded with quercetin using the layer-by-layer technique, and determined that the chemical structure of the layers is an important factor in obtaining antioxidant activity in the multilayer coating [105]. 

In Table 2, the bioactive properties of CG and its various applications in the biomedical field are summarized. 

In general terms, we can mention that the bioactive properties of CG are very important for the development of new therapeutic agents, capable of presenting high efficacy and safety, which allow them to be used at some point in the world population. At this time, we are at a crucial point in the development and research of new therapeutic agents derived from seaweed, since in all the studies analyzed, it has been seen that carrageenans can act by themselves, as bioactive agents or can improve their capacity bioactive by binding to drugs or by transforming or modifying their chemical structure. This transformation or modification of the original sulfated polysaccharides into new ones allows enhanced bioactive properties to be obtained that improve treatments or prevent diseases. 

## 4. Carrageenan in Biomedical Applications 

### 4.1. Applications of Carrageenan in Drug Delivery Systems

Initially, excipients were used almost exclusively as components that contributed to the manufacturing processes of the pharmaceutical forms, but, thanks to the boom in the obtaining, purification, and use of biodegradable, biocompatible, and non-toxic compounds in pharmaceutical formulations, it led to their transformation into multifunctional components, which can provide bioactive and functional properties in drug development. Hence, the biomedical importance of CG as bioactive components that make it possible to improve pharmaceutical formulations and in certain cases be adjuncts to other active principles. The biological and chemical properties of CG are the main reasons they are used in drug delivery systems [40,106]. 

The chemical structure of CG is a factor that explains the increase in its applications in drug delivery systems, since it has three important characteristics [40,107]: (a) its glycosidic bonds allow it to be cleaved by hydrolase enzymes, producing biodegradability; (b) the sulphate groups in the CG are anionic and enhance the behavior of polyelectrolytes; and (c) the presence of hydroxyl groups provides the necessary interactions to produce chemical modifications.

The major applications in the field of biomedicine are shown as a schematic layout in Figure 6. CGs are applied in various pharmaceutical formulations, including tablets [106,108], suppositories [109,110], films [111], fast-dissolving inserts (FDIs) [63], beads [112,113], pellets [114,115,116], microparticles [117,118,119], nanoparticles [120,121,122,123,124,125], inhalable systems, injectables [126], and hydrogels [127,128,129]. In addition, recent studies have shown that CGs are promising candidates in tissue engineering, thanks to their similarity to native glycosaminoglycans [127,128,130,131]. 

CGs perform various functions in these pharmaceutical formulations, ranging from the formation of matrices, stabilizers, binders, disintegrators, solubilizers, thickeners, and coatings, to more complex processes, such as drug release control. Often, they do not fulfil a single function but are cumulative. CG readily form a gel, so they are commonly used in drug delivery systems, generally formed through heat-reversible gelation, ionic crosslinking, and the modification of the carrageenan main chain. The chemical structure determines their functional properties, stability, biodegradability, and biocompatibility. CGs are used in the pharmaceutical formulations described below.

#### 4.1.1. Carrageenan-Based Tablets 

Hydrophilic matrix systems are the first drug delivery systems applied to control drug release. Due to the physicochemical properties of CG, they are used as matrices for loading drugs with low bioavailability. The potential of *iota*-CG and *lambda*-CG in the preparation of controlled-release tablets was investigated. Theophylline, sodium salicylate, and chlorpheniramine maleate were used as model drugs. Matrices of 500 mg were used, with a diameter of 1/2”. When they studied the effect of the drug to CG ratio and the diameter of the tablet on the release profile, they used matrices of two different ratios of drug to CG―13.5% and 16.9%―and two different diameters (3/8”and 1/2”). The results of the study showed that the matrices that contained CG were useful to produce controlled release between 8 and 12 h, with release profiles approaching a zero-order kinetic [133]. Tablet diameter, drug to CG ratio, and the ionic strength of the dissolution media appear to play a role in drug release; hence, the release rate increased both with the tablet diameter and a higher drug-to-CG ratio. These factors should be considered when designing sustained release formulations with CG [133].

Buchholz et al. reported modified-release matrix tablets based on *iota*-CG combined with microcrystalline cellulose and lactose in different ratios, containing riboflavin 50-phosphate sodium (vitamin B_2_) as the active pharmaceutical ingredient. The dissolution study was tested in gastric fluid (pH 1.2) and phosphate buffer (pH 4.5), and the results confirmed the formation of controlled-release tablets. The release profile showed a uniform increase of up to 80%. Lactose influenced the behavior of the matrix; when there is less lactose, the elasticity of the matrix is greater, resulting in tablets with weak mechanical properties [134].

*Lambda*-CG and two soluble model drugs (diltiazem HCl or metoprolol tartrate) were evaluated in matrix tablets. The two complexes studied released the drug through a different mechanism, indicating two distinct drug/polymer interaction strengths. Diltiazem/CG produced a poorly soluble slow-dissolving matrix, while metoprolol tartrate/CG was a less stable complex, resulting in an erodible matrix in only 8 h. The authors conclude that the study suggests two distinct drug delivery systems: metoprolol tartrate follows a classical soluble matrix-type delivery system, whereas in diltiazem HCl, the dissolving/diffusing species is the complex itself due to the very strong interaction between the drug and the polymer. In this case, the tablet must be considered a monolith constituted by a single insoluble compound rather than a matrix system [135].

Another study reported the mechanism of release of a cationic drug (doxazosin) from tablets based on *lambda*-CG. The experiments were carried out in solutions with different ionic strengths and with the addition of an anionic surfactant, sodium dodecyl sulphate (SDS). The results showed that the release rate depends strongly on cooperative drug/polymer interactions. The addition of SDS at concentrations below its critical micelle concentration (CMC) slows drug release through the hydrophobic binding of SDS to the drug/polymer complex, while at concentrations over the CMC, SDS draws drug from the complex by forming mixed micelles with it, thus accelerating the release [136].

Yermak et al. evaluated different types of CG as matrices that include echinochrome A (Ech) in order to protect and improve drug dissolution. The inclusion of Ech in complexes with CG decreased its oxidative degradation, improved Ech solubility, and caused changes in the morphology, charge, and size of the CG; the CG/Ech complex also exhibited mucoadhesive properties. The release rate of the Ech depended on the structure of the polysaccharide and the specific ions. The authors indicate that the activity of the complexes exceeded the activity of the reference drug, so new CG-based matrices may be beneficial for oral administration and prolong the action of Ech [137].

New applications have also been explored, including vaginal tablets, which are pharmaceutical forms that have so far been little studied. *Kappa*-CG-based vaginal tablets were investigated for acyclovir-controlled release. Drug release studies were carried out in two ways: in simulated vaginal fluid and in a simulated vaginal fluid/simulated seminal fluid mixture. Bioadhesive capacity, bioadhesion residence time, and in vitro biocompatibility were also evaluated. The tablets resulted in a prolonged release of the drug for a period of seven days. It was demonstrated that the combination of *kappa*-CG with hydroxypropyl methylcellulose (HPMC) was crucial for sustained release. The mucoadhesive capacity of the polymers and the residence time of the *kappa*-CG/HPMC formulations was between 72 and 108 h. In terms of biocompatibility, the results showed that *kappa*-CG had no cytotoxicity at the maximum concentration tested and might be beneficial in the prevention of genital herpes infection [108].

In a more recent study, vaginal tablets based on *iota*-CG and HPMC were developed and evaluated. The combination of *iota*-CG and HPMC achieved the controlled release of acyclovir in 96 h. These vaginal tablets were indicated for helping prevent the sexual transmission of genital herpes, as the microstructure formed by the polymer mixture has an adequate swelling rate and high mucoadhesive capacity, which allows the formulation to remain in the vaginal area long enough to ensure the complete release of the acyclovir (Figure 7) [106]. 

#### 4.1.2. Carrageenan-Based Suppositories 

Microbicide research is currently booming due to its advantages in preventing sexually transmitted infections. Semisoft vaginal suppositories have been developed from *kappa*-CG using tenofovir (TFV) as an antiviral drug. In vitro dissolution studies in water, vaginal simulant fluid, and semen simulant fluid demonstrated that at least 45–50% of the TFV diffuses out of the suppositories within the first two hours in both fluids, irrespective of size and volume [109]. 

In a recent study, Zaveri et al. developed semisoft suppositories from mixed polymer combinations of CG and Carbopol^®^ 940P with TFV. The results show that between 45% and 50% of the TFV was released in the first 2 h, 60% in 6 h, and 70% in 24 h. There were no significant differences in the initial TFV release rate of the various gel combinations of carrageenan and Carbopol^®^, or in the amount of TFV released over 48 h. The authors determined that the formulations containing more Carbopol^®^ had a better acceptability rate; that the shape, size, firmness, and properties of the product after insertion influence the use of these microbicides; and that the combination of *kappa*-CG with Carbopol^®^ substantially improves its acceptability [110].

#### 4.1.3. Carrageenan-Based Fast-Dissolving Insert 

A fast-dissolving insert was developed to test its effectiveness against HSV-2. The studies began by combining CG with griffithsin (GRFT). In vivo studies demonstrated that the combination of GRFT with CG in a freeze-dried fast-dissolving insert (FDI) formulation for on-demand use protects rhesus macaques from a high dose of vaginal SHIV SF162P3 challenge 4 h after application. The formulation also protects mice vaginally against HSV-2 and HPV pseudovirus. The researchers determined that the formulation is a safe, potent, broad-spectrum, and on-demand non-antiretroviral product, and that GRFT/CG FDI warrants subsequent clinical development. This study shows that CG is not only used as an excipient but also as a bioactive agent. In addition, GRFT or CG are not easily absorbed after topical administration, which makes these antiviral agents ideal for repeated/extended topical use [63].

#### 4.1.4. Carrageenan-Based Beads 

CG-based beads were also described as possible drug carrier systems. The development of sustained-release mefenamic acid beads, a non-steroid anti-inflammatory drug based on *kappa*-CG, was reported, aimed at reducing the daily dose and decreasing the gastrointestinal disorders caused by traditional medicine [138]. 

CG combined with chitosan forms a polyelectrolyte complex and has been studied as controlled release systems of diclofenac sodium. The results showed that the release rate of diclofenac can be controlled over long periods by using 2:1 chitosan:carrageenan with 5% (*w*/*v*) drug. The beads crosslinked with glutaric acid showed better results than the non-crosslinked beads, and the beads crosslinked with glutaraldehyde were most effective for the prolonged release of the drug over 24 h. It was also observed that the release of diclofenac is slower in pH 1.2 and pH 6.8 and much faster in pH 7.4, offering an effective controlled-release system in the administration of specific colon-targeted drugs [139].

Other research focused on pH-sensitive hydrogel beads based on polyacrylamide grafted on *kappa*-CG and sodium alginate to target ketoprofen to the intestine. The beads exhibited ample pH-responsive behavior in the pulsatile swelling study. The ketoprofen release was significantly increased when the pH of the medium was changed from acid to alkaline. The beads showed a maximum of 10% drug release in acidic medium of pH 1.2, and about 90% drug release was recorded in alkaline medium of pH 7.4. The stomach histopathology of albino rats indicated that gastric side effects, such as ulceration, hemorrhage, and erosion of the gastric mucosa, were reduced when the drug was loaded into pH-responsive hydrogel beads [113].

The preparation of magnetic and pH-sensitive beads based on *kappa*-CG and sodium alginate for use as drug-targeting carriers has also been described. This research studied the swelling behavior of beads as a function of pH, and found that the sensitivity of hydrogels to the pH of the media was lower as the *kappa*-CG ratio increased. Riboflavin was loaded in hydrogel beads and the drug release profiles indicated pH-dependent behavior with high release at pH 7.4. The lowest release content at acidic pH was observed for beads containing only an alginate component, a behavior that originated from the pH sensitivity of the alginate biopolymer [112]. In a later study, the same authors developed magnetic carboxymethyl chitosan/*kappa*-CG beads for drug delivery, using diclofenac sodium as a model system. The results of the in vitro drug release from the beads revealed a pH-dependent behavior. The content of released drug was found to be low at pH 1.2, while at pH 7.4, the diclofenac sodium content released was considerably enhanced for all beads. The researchers concluded that the presence of magnetite nanoparticles undoubtedly influenced the drug release patterns, and the beads’ response to external stimulus makes them good candidates for novel drug delivery systems [140]. 

A formulation of solid lipid pearls based on CG showed the potential to mask the bitter taste of enrofloxacin and prolong its release rate [141]. Another study on alginate-pectin microbeads encapsulating caffeine indicated that the use of natural biopolymers, such as CG, chitosan, and psyllium, as wall materials for the encapsulation of caffeine may positively modify the perception of bitterness [142].

Bio-nanocomposite hydrogel beads based on *kappa*-CG and bio-synthesized silver nanoparticles (Ag-NPs) were analyzed as an antimicrobial agent. *Kappa*-CG/Ag-NPs presented good antibacterial activities against *S. aureus*, methicillin-resistant *S. aureus*, *P. aeruginosa*, and *E. coli*, with maximum zones of inhibition of 11 ± 2 mm. The cytotoxicity results showed that *kappa*-CG/Ag-NPs bio-nanocomposite hydrogels have an acceptable level of toxicity and great pharmacological potential due to their suitable level of safety for use in the biological systems [143]. 

*Kappa*-CG-based beads have also been developed as vehicles for administering curcumin (Cur) in cancer cells, achieving drug release under pH-dependent conditions (5.0 and 7.4) (Figure 8). Cur was effectively incorporated into *kappa*-CG, which has been found to have an enhanced ability to efficiently load and release Cur in a well-defined medium for controlled release. The controlled release of Cur occurred at pH 5.0 through in vitro studies. The cytotoxicity of the Cur-loaded *kappa*-CG had a significantly high apoptotic activity in selected lung cancer cells of A549. The authors suggested that the Cur-loaded *kappa*-CG could be a potential candidate for developing polymer-based natural delivery vehicles with an improved therapeutic function for pharmaceutical applications [144]. 

#### 4.1.5. Carrageenan-Based Pellets 

Various reports have shown that *kappa*-CG is a novel pelletization aid with high formulation and process robustness [114,145,146,147]. Pellets made with *kappa*-CG have the advantage of rapid disintegration compared to others, such as microcrystalline cellulose (MCC) pellets, which is associated with rapid drug release, particularly in the case of poorly soluble active ingredients [146]. The rapid drug release from *kappa*-CG pellets may be beneficial in formulating enteric-coated pellets by this method. Ghanam et al. investigated the feasibility of formulating multiparticulate tablets with coated *kappa*-CG pellets; the results showed that *kappa*-CG granules are more advantageous than MCC granules for the formulation of multiparticulate tablets with enteric properties. *Kappa*-CG granules could also be formulated in multiparticulate tablets with sustained release properties [114].

In another investigation, nifedipine granules were developed using HPMC K15 M, *kappa*-CG, and MCC as a sustained mucoadhesive delivery system. The results indicate that *kappa*-CG has promising potential for use in the production of pellets with the desired size, sphericity, flow, and hydrogel-forming abilities, and robust formulation development. *Kappa*-CG, MCC, and HPMC K15 M at a ratio of 20:35:10 *w*/*w* could serve as an effective carrier for enhancing the sphericity and sustained release of matrix pellets, demonstrating that the combination with other polymers improves the formulation conditions [148].

A recent study sought to develop pellets without MCC in order to avoid cases of drug incompatibility or the lack of disintegration of the granule matrix. The authors investigated a combination of *kappa*-CG as a spheronization aid, chitosan as a diluent, and Carbopol^®^ 974P as a binder in the production of pellets, using acetaminophen as a model drug. The results of the screening design demonstrated the feasibility of producing high-quality pellets without MCC. The pellets were spherical and robust and met the criterion of immediate release [115].

#### 4.1.6. Carrageenan-Based Films 

Polymeric films are increasingly being investigated as a promising pharmaceutical form for drug delivery and wound dressings due to advantages, such as their portability, application form, retention time, and easy storage and stability. These films can be used for immediate or controlled release depending on the polymer composition of the formulation [149,150].

Polyethylene oxide (Polyox) and *kappa*-CG-based solvent cast films were formulated as dressings for drug delivery of streptomycin and diclofenac. The drug-loaded films showed a high capacity to absorb simulated wound fluid and significant mucoadhesion force, and also allowed the controlled release of both streptomycin and diclofenac for 72 h. These films produced higher zones of inhibition against *S. aureus*, *Pseudomonas aeruginosa*, and *E. coli* compared to the zones of inhibition of the individual drugs. The dressing can help to reduce bacterial infection due to the antimicrobial action of streptomycin, and potentially in synergy with diclofenac; while the latter can also help reduce the inflammation and pain associated with injury thanks to its anti-inflammatory action [151].

Shojaee-Aliabadi et al. developed a composite film based on CG and two essential oils—Zataria multiflora Boiss (ZEO) and Mentha pulegium (MEO)—to assess the film’s antimicrobial effect and antioxidant activity. The films containing essential oils showed good antioxidant properties, and this effect was greatly improved by the addition of ZEO. Films containing ZEO showed significant antimicrobial effects. The results suggest that, as natural antioxidants and antimicrobials, essential oils have potential for use in CG film while producing some changes in mechanical and optical properties [152]. 

In other research, Fouda et al. reported two types of composite films. The first comprised *kappa*-CG, polyvinyl pyrrolidone (PVP), and polyethylene glycol (PEG), and the second consisted of *kappa*-CG, PVP, and PEG trapped with biosynthesized Ag-NPs. The results showed that Ag-NPs induced significant hydrophilicity in the composite film with a contact angle of 34.7°. The TGA results also revealed higher stability for the film with Ag-NPs and higher strength properties than other films without. A low swelling property was observed with composite films containing Ag-NPs. The advanced microstructure analysis also proved the in situ deposition of Ag-NPs on the surface of the composite film. Ag-NPs alone and composite film showed higher fungal activity against *Aspergillus* spp., *Pencillin* spp., and *Fusarium oxysporum* compared to fluconazole. The composite film with Ag-NPs can be used in biomedical applications [153].

Gu et al. developed an intravaginal film platform for targeted delivery of small interfering RNA (siRNA)-loaded nanoparticles (NPs) to dendritic cells as a potential gene therapy for the prevention of sexually transmitted human immunodeficiency virus (HIV) infection. The polymer film was developed with PVA (polyvinyl alcohol) and *lambda*-CG loaded with the nanoparticles. The polymer ratio influences the appearance and disintegration time of the formulation; the PVA concentration increased the films became stiff and opaque. In general, this application demonstrates that the use of carrageenan improves the condition of the film and allows its application and administration by the vaginal route. This study establishes a basis for the development of films formed by carrageenan or in combination for loading drugs for vaginal administration [111].

Carrageenan-based hydrogels and dried hydrogel films were prepared using KCl as a crosslinker and zinc oxide metallic nanoparticles and copper oxide nanoparticles as a functional filler. The effects of nanofillers on the morphology, swelling, and mechanical, thermal, optical, and antibacterial properties of the hydrogel were tested. The properties of carrageenan-based nano-hydrogels were evaluated with both hydrogel and dried film forms. The results indicated that the carrageenan-based nanocomposite hydrogel films had stronger antibacterial activity against *E. coli* than *Listeria monocytogenes*. Carrageenan-based nanocomposite hydrogels and dried films have a high potential for use as functional skincare products, such as facial masks, and as a wound dressing [154].

In other research, some types of ultratough polysaccharide-based hydrogel films prepared by the complexation of *kappa*-CG and chitosan (CS) in a wide range of weight/charge ratios were evaluated. The gel films were prepared by casting the mixture of oppositely charged *kappa*-CG and protonated CS solutions and subsequently swelling the dry film in water to achieve the equilibrium state (Figure 9). A series of polysaccharide-based hydrogel films were obtained with notable mechanical performance and good cell antiadhesion property. The hydrogel films obtained had a water content of 48–88% and a thickness of 40–60 μm, and revealed good mechanical performance with a tensile stress at break (σb) of 2–6.7 MPa, an elongation at break (εb) of 80–120%, and a Young’s modulus (*E*) of 1.2–25 MPa. These hydrogel films also had good biocompatibility and cell antiadhesion properties, which are crucial for their application as an artificial dura mater and diaphragm materials during surgery [155].

In recent research, Jaiswal et al. prepared a *kappa*-CG-based hydrogel film for healing wounds, incorporating chitosan-capped sulphur nanoparticles (SNPs) and grapefruit seed extract. The combined hydrogel film showed higher mechanical strength, swelling ratio, and ultraviolet barrier properties than the carrageenan film. It had strong antibacterial activity against *Staphylococcus epidermis* and *E. coli* and high biocompatibility against mouse fibroblasts. The *kappa*-CG/grapefruit seed extract/SNP 3% hydrogel film showed full-thickness wound healing efficacy in Sprague-Dawley rats in a short time. Histological examination showed the complete appearance of the healed epidermis. These hydrogel films could be used for the treatment of full-thickness wounds [156].

#### 4.1.7. Carrageenan-Based Oral Suspensions 

Drugs with CG ionic interaction can result in lower drug solubility and dissolution retardation, which can be used to develop sustained-release dosage forms of water-soluble drugs [157]. 

In this context, Bani-Jacer et al. combined ambroxol-CG complexation and raft formation to develop a sustained-release liquid suspension of ambroxol (ABX). The complex was formulated as suspensions in an aqueous raft-forming vehicle of sodium alginate and calcium carbonate. Suspensions containing ABX-CG complex or ABX showed rapid raft formation and raft floatation in the dissolution medium. The suspensions of ABX-CG complex showed much more control over ABX release than free drug suspensions. The drug release from the complex suspensions was biphasic with an initial burst release followed by a sustained release phase with a very slow drug release rate. This sustained release of complex suspensions over free drug suspensions can be attributed to the fact that one more step was required for the release prior to drug diffusion, namely the dissociation of the ABX-CG complex. The authors noted that the biphasic release of ABX could be advantageous to treat acute and chronic respiratory diseases accompanied by increased mucus production, as rapid relief from the accompanying annoying symptoms is a priority. If the relief is maintained for a prolonged time, a lower dosage frequency would be required [157]. 

#### 4.1.8. Carrageenan-Based Micro/Nanoparticles 

The development of CG-based nanoparticles has also been addressed by many authors [158]. In recent years, nanoparticles based on the CG/chitosan combination for drug administration have been extensively investigated. They are generally prepared by the methods of ionic gelation or the formation of polyelectrolyte complexes (PEC) by mixing CG with cationic polymers [159].

The potential ability of *kappa*-CG, *iota*-CG, *lambda*-CG, and chitosan (CS) to form a controlled-release system for glucose oxidase (GOD) was reported. CS/CG complexes at charge ratios of 3 and 5 can entrap GOD and control its release. Electrostatic interaction is the main mechanism involved in incorporating the protein into the complex, and the controlled release of GOD from the complex is due to the swelling mechanism of the CS/CG complex. The authors point out that CS/CG complexes are a promising drug delivery system for oral administration of peptides and proteins [160].

Grenha et al. described the preparation and characterization of protein-loaded nanoparticles obtained by ionic complexation of CS/CG as a drug delivery system. The in vitro release assay demonstrated that the CS/CG nanoparticles provided a sustained and controlled release of the protein for three weeks. The nanocarriers also exhibited low toxicity in contact with fibroblast-like cells, which is an encouraging indicator of their biocompatibility and safety [161].

CS/CG/tripolyphosphate (CS/CG/TPP) nanoparticles were prepared by polyelectrolyte complexation/ionic gelation. Tripolyphosphate (TTP) acted as a crosslinker agent and enabled the production of nanoparticles with a smaller size, in addition to enhancing the production yield. The presence of TPP in the nanoparticle matrix also increased their stability to at least nine months. The researchers demonstrated that charge ratios play a critical role in the formation of nanoparticles, since a ratio of around 1 leads to precipitation, owing to charge neutralization, while very high charge ratios do not provide enough charges to allow an interaction that induces nanoparticle formation. These nanoparticles are potential candidates for application in the mucosal delivery of macromolecules due to their small size and positive load [162]. In further work, they reported the suitability of CS/CG/TPP nanoparticles as carriers for an application in pulmonary and nasal drug delivery. The nanoparticles proved to be stable in the presence of an amount of lysozyme simulating in vivo conditions. The carriers exhibited low or negligible toxicity in contact with two respiratory cell lines representing both the nasal and pulmonary epithelium, as demonstrated by the effects on cell metabolic activity and transepithelial electrical resistance [163].

Rosas-Durazno et al. described a prototype of nanocapsules based on the electrostatic interaction of oppositely charged *kappa*-CG polyelectrolyte with the surface of a dodecyl-trimethylammonium chloride (DTAC)-stabilized nanoemulsion. The authors noted that the introduction of *kappa*-CG at the surface of a nanoemulsion could offer the possibility to exploit the thermoreversible coil-to-helix transition characteristic of *kappa*-CG and other related galactan polysaccharides as a strategy to develop temperature-sensitive nanocarriers. The addition of negatively charged polyelectrolyte to the surface of a nanoemulsion stabilized by DTAC could also contribute to reducing its potential cytotoxicity [164]. 

Curcumin-loaded nano-microparticulate systems to improve the solubility and stability of curcumin in gastrointestinal conditions were reported. These nano-microparticle carriers are based on CS, CG, and alginate. The results showed that over 95% of the loaded curcumin was released after 7 h of incubation in a pH 7.4 buffer solution using freeze-dried microparticles with an alginate/CG ratio of 50:50. CG plays an important role in improving the release pattern of curcumin from the hydrogel matrices, as higher release was observed in formulations with a higher CG content [165].

Lectin-functionalized carboxymethylated *kappa*-CG microparticles with entrapped insulin were evaluated in in vitro and in vivo studies. The insulin entrapped in lectin-functionalized carboxymethylated *kappa*-CG microparticles was protected from hydrolysis and proteolysis by stomach acids and enzymes, and the oral administration of the insulin entrapped in the microparticles led to the prolonged duration of the hypoglycemic effect for up to 12–24 h in diabetic rats. The authors concluded that these lectin-functionalized carboxymethylated *kappa*-CG microparticles have the potential for the development of an oral insulin delivery system [166].

Polyelectrolyte nanocapsules of *kappa*-CG and CS polyelectrolyte complex were prepared by encapsulating neem seed oil (NSO). Nanocapsules were freed from NSO and then loaded with isoniazid. The drug was initially loaded at a rapid rate and finally at a slower rate. The release rate of isoniazid was dependent on the pH of the medium, and was higher in acidic than in basic media [167].

Bosio et al. reported a hybrid nanoporous microparticle (hNP) carrier based on calcium carbonate and biopolymers derivatized with folic acid (FA) and containing doxorubicin as a chemotherapeutic drug model. The coupling of *lambda*-CG to FA in the microparticles (FA–*lambda*-CG-hNPs) increased the cancer-cell targeting and also extended the specific surface area. A sustained release was found under simulated physiological conditions for *lambda*-CG and FA–*lambda*-CG hNPs in 25-day experiments. FA–*lambda*-CG-hNP anticancer activity on the human osteosarcoma MG-63 cell line showed cell viabilities of 13% and 100% with and without doxorubicin, respectively; this formulation is a novel platform for cancer chemotherapy [168].

In another study with *iota*-CG, γ–maghemite (γ-Fe_2_O_3_) nanoparticles were combined with *iota*-CG to develop a novel nanocomposite material. The results confirm the integration of maghemite nanoparticles to the sulphate groups of CG. These nanoparticles have properties that can make them attractive in biomedical applications. Tuning the surface properties of this nanocomposite by changing the concentration ratio of *iota*-CG and γ-Fe_2_O_3_ can make them more dynamic in targeting cancer cells. The ability to inhibit growth in cancer cells without being cytotoxic to normal cell lines makes it a promising nanovector in drug delivery [169]. 

Magnetic CS/*kappa*-CG carriers as a promising pH-responsive smart drug delivery system for sunitinib release were prepared. Magnetic carriers had a good rate of drug incorporation and allowed the sustained release of sunitinib. The release of sunitinib from all nanoparticles was completed in seven days. The researchers reported that magnetic carriers could be used in cancer therapy, as the pH-dependent release of sunitinib content is lowest at pH = 7.4, so magnetic carriers may be promising candidates for anticancer drugs with reduced side effects [170]. 

In a recent study, *kappa*-CG-wrapped ZnO nanoparticles were designed and evaluated as antibacterial agents against infectious diseases caused by microorganisms resistant to multiple drugs (MRSA). The nanoparticles significantly inhibited the growth of the “MRSA superbug” and the formation of biofilms at a low concentration, and also reduced the hydrophobicity of the bacterial cell surface. No hemolysis and no morphological change in human red blood cells were recorded. The nanoparticles were not toxic to NIH3T3 mouse embryonic fibroblast cells and no changes in viability were observed up to 500 μg/mL. The ecosafety assessment with *Artemia salina* showed that the nanoparticles were non-toxic to the environment. This study offers a “nano-anti-superbug drug” with broad-spectrum activities that may be a good candidate as a nano-antibiotic to treat MRSA infectious diseases [122].

Polysaccharide-based aerogels combine the properties of aerogel and polysaccharides, which allows them to be tailored to several applications. In this context, Alnaief et al. reported a microparticle aerogel using a biocompatible polymer. Microspherical carrageenan gel particles were obtained by applying emulsion technology. The gel was converted to an aerogel using a supercritical carbon dioxide extraction process. The results indicated that the surface area of the aerogel ranged between 33 and 174 m^2^/g, and the average pore volume and pore size were 0.35 ± 0.11 cm^3^/g and 12.34 ± 3.24, respectively. The porous material thus shows potential characteristics for its application for drug delivery [171].

In another investigation, Obaidat et al. developed CG spherical aerogel microparticles as a potential drug carrier. Ibuprofen was selected as a model drug and the formulation was prepared using the emulsion-gelation technique. The prepared CG aerogel microparticles showed significant potential for use as a drug carrier, as ibuprofen was successfully loaded in the amorphous form inside the prepared microparticles with a notable enhancement in the drug release profile [118].

#### 4.1.9. Carrageenan-Based Floating System

*Kappa*-CG-based floating hydrogel has currently gained widespread attention among researchers. By incorporating carbonate and bicarbonate salt, hydrogels can be made to constantly float in the stomach and deliver drug in a controlled manner. Selvakumaran and coworkers prepared genipin crosslinked floating *kappa*-CG hydrogel and investigated the effects of genipin on the physical, chemical, and mechanical properties of hydrogel and the in vitro release of ranitidine hydrochloride. The result showed that all the hydrogels formulated had excellent floating behavior, and that the crosslinking reaction exerted a significant effect on gel strength, porosity, and swelling ratio compared to non-crosslinked hydrogels. It was found that the drug release was slower and reduced after being crosslinked [172].

In additional work, floating hydrogels were prepared from *kappa*-CG containing CaCO_3_ and NaHCO_3_ as pore-forming agents. Amoxicillin trihydrate was used as a model drug. The incorporation of CaCO_3_ into *kappa*-CG hydrogel showed smoother surface gels, higher drug entrapment efficiency, and a sustained drug release profile due to its low porosity compared to those produced with NaHCO_3_ [173]. 

#### 4.1.10. Carrageenan-Based Intranasal System 

*Iota*-CG demonstrated antiviral activity against human rhinoviruses and the influenza A virus when it was applied via the nasal route, and several studies mentioned in the antiviral activity section have been developed [66,67,69,70]. The non-clinical safety of intranasal *iota*-CG was evaluated. The animal experiments included repeated dose-local tolerance and toxicity studies with intranasally applied 0.12% *iota*-CG for 7 or 28 days in New Zealand White rabbits and nebulized 0.12% *iota*-CG administered to F344 rats for 7 days. The results show no evidence of local intolerance or toxicity when CG was applied intranasally or by inhalation. The authors thus conclude that 0.12% of *iota*-CG is safe for clinical use via intranasal application [174]. The safety assessment of *iota*-CG has allowed the development of new nasal application formulations. 

Nasal inserts composed of polyelectrolyte complexes (PECs) based on *kappa*-CG and CS to boost the therapeutic efficacy of sumatriptan succinate were reported. The polyanion/polycation molar ratio plays a critical role in modulating the characteristics of the inserts. *Kappa*-CG/CS (4:1) demonstrated the highest water uptake ability and mucoadhesive potential and provided a more controlled release of sumatriptan succinate, pointing to this formulation as a promising delivery system for sumatriptan succinatein in the treatment of migraine attacks [175].

#### 4.1.11. Carrageenan-Based Wafers 

Kianfar et al. developed a buccal wafer prepared by freeze-drying gels based on *kappa*-CG, pluronic acid and polyethylene glycol 600 loaded with model soluble (paracetamol) and insoluble (ibuprofen) drugs for buccal delivery. Buccal wafers were obtained by gels combining two polymers: *kappa*-CG 2% (*w*/*w*) and pluronic acid 4% (*w*/*w*) with PEG 600 4.4% (*w*/*w*) and 0.8% (*w*/*w*) or 1.8% (*w*/*w*) ibuprofen and paracetamol. The authors conclude that the wafers showed ideal release patterns in conditions simulating those of saliva, and that coupled with their desirable mucoadhesive characteristics have potential for buccal drug delivery [176].

In another study, Pawar et al. evaluated wafers based on polyox with CG or sodium alginate containing streptomycin and diclofenac to improve chronic wound healing. The wafers showed controlled release of streptomycin and diclofenac due to differences in pore size, and the formation of sodium sulphate due to the salt forms of the two drugs. The researchers mentioned that the wafers may potentially be used for highly exuding wounds, such as chronic ulcers [177]. 

#### 4.1.12. Carrageenan-Based Hydrogel 

Hydrogels are three-dimensional hydrophilic polymer networks that can absorb and retain large quantities of water, saline, or physiological solutions [178]. Hydrogels based on natural biopolymers, especially polysaccharides, have drawn considerable attention due to their biodegradability, biocompatibility, renewability, and safety, compared to synthetic polymer-based hydrogels. Gels are the most characteristic dosage forms of CG, which has allowed their exploitation in various areas, mainly as drug delivery, thanks to their strong gel-forming ability and high water-holding capacity [154,179,180]. 

The effect of spherical and rod-shaped Au nanoparticles (NPs) in the microstructure, and the thermomechanical and release properties of thermosensitive *kappa*-CG hydrogels were investigated. Thermal and mechanical analyses of the composites revealed that Au NPs reinforce the hydrogel structure. The effect of the nanoparticles on the microstructure and the strength of the hydrogel had implications on the controlled-release mechanism as demonstrated by in vitro release studies using a substance model (methylene blue). *Kappa*-CG hydrogels containing Au NPs exhibited not only optical features modulated by the morphology of the fillers but also behavior as drug carriers that can be adjusted by the characteristics of the Au NPs [181].

Another study reported the synthesis and characterization of a biodegradable silver nanocomposite hydrogel based on *iota*-CG as a potential antibacterial agent. The results showed that the silver nanocomposite hydrogels developed were effective as potential candidates for antimicrobial applications [182].

Modifications to the physical properties of CG have been developed for the controlled release of drugs, for example, with the addition of genipin. This modification produced the relaxation of the CG molecules and the swelling mechanism, giving rise to the controlled release of B-carotene when using genipin [183]. In other research, *kappa*-CG/polyvinyl alcohol crosslinked hydrogels were formulated using genipin as a natural and non-toxic crosslinker to achieve controlled drug release. The results showed that using genipin can stop burst release in hydrogels and control active material better than native gels as a result of their structural modification. The morphological changes in the polymer network caused by changing the crosslinking ratio also contributed to eliminating the burst effect. This suggests that the burst release is highly dependent on the degree of crosslinking and the mesh space available for drug diffusion [184]. 

A hydrogel membrane constituted of *kappa*-CG and hyaluronic acid (HA) crosslinked with epichlorohydrine was evaluated. All the prepared membranes were loaded with l-carnosine as a drug model. The swelling properties of all hydrogel membranes showed impartial dependency on the high presence of hyaluronic acid; the cumulative release profile rose significantly to between 28% and 93% as HA content increased. A slight change in tensile strength occurred by increasing the HA% to *kappa*-CG, while the highest value of strain for the *kappa*-CG membrane was 498.38% using 3% HA. The thermal stability of the *kappa*-CG/HA was higher than that of HA. The authors concluded that this hydrogel membrane can be used in a wide range of biomedical applications [185].

Mahdavinia and coworkers evaluated a sustained release of ciprofloxacin using CS/hydroxyapatite/*kappa*-CG complexes. The ciprofloxacin-loaded hydrogel nanocomposites exhibited antibacterial activity against Gram-positive *S. aureus* and Gram-negative *E. coli* bacteria. Due to the hydroxyapatite introduced, the release of ciprofloxacin occurred in a sustained manner. While the pristine CS/*kappa*-CG complex released about 98% of ciprofloxacin over 120 h, only 52% and 66% of the loaded drug was released from hydrogel nanocomposites containing a high and low content of hydroxyapatite, respectively. The sustained release of ciprofloxacin from the hydrogel nanocomposites identifies them as a potential candidate for designing drug delivery systems with prolonged release ability [125]. 

Rassol et al. developed carrageenan-based stimuli-responsive hydrogels with PVA and a silane crosslinker. All the hydrogel samples showed strong antibacterial activity against *S. aureus* and some activity against *E. coli*. The in vitro release of cephradrine increased with time and with rising pH; 85.5% of the drug was released in a controlled manner in 7.5 h. The researchers indicated that these hydrogel samples could be an appropriate candidate for injectable drug delivery [81]. 

In a recent study, the authors evaluated a pH-sensitive hydrogel based on CG, sodium alginate, and PEG crosslinked with (3-aminopropyl) triethoxysilane, demonstrating that the hydrogel produces the controlled release of lidocaine and good cell compatibility. The prepared hydrogels can be applied as a successful drug delivery system for controlled release [186].

Among other applications, it is worth mentioning a recent study in which an oral controlled-release carbamazepine gel is formulated for pediatric use. In this work, an *iota*-CG gel loaded with carbamazepine alginate beads was developed, and the appearance, syneresis, uniformity of the drug content, rheology, release profile, and stability of the final gel were evaluated. The gel showed good homogeneity, good shear thinning properties, and no syneresis. The release profile of the beads and the gel were not significantly affected and remained similar to the control formulation. Finally, the results of the physical stability studies showed that the gel formulation maintained its properties for a month when stored at 30 °C. The authors found that carbamazepine in a gel-sustained release dosage form combines the advantages of the suspension form in terms of dosing flexibility, and the advantages of the tablet form in regard to the sustained release profile [187].

In another study, a composite hydrogel was prepared as a dual drug delivery carrier. Microparticles of ketoprofen and mupirocin were embedded in the *kappa*-CG/locust bean gum hydrogel. For both drugs, the hydrogel showed a slower release than from the microparticles and hydrogel separately, reaching a period of 7 days at 37 °C. The release was drug controlled thanks to the combined microparticles and hydrogel. This system can therefore be considered as a possible carrier of low water-soluble drugs, mainly for wound-healing applications [117].

A final study indicated that *kappa*-CG had the potential to improve the physicochemical properties and drug release performance of agar hydrogel. A series of agar/*kappa*-CG mixed hydrogels with different mass ratios were analyzed. The results showed that the gel strength, gelling temperature, and gel melting temperature decreased as the CG increased, and the apparent viscosities increased. These hydrogels were also used as carriers for the delivery of metformin hydrochloride. The drug-loading efficiency and the sustained release capacity of agar hydrogels could be enhanced by the addition of *kappa*-CG until 9 h. It should be noted that the release profile was mainly dominated by the electrostatic interaction between the drug and the polysaccharides [188].

### 4.2. Carrageenan-Based Tissue Engineering

#### 4.2.1. Bone or Cartilage 

Carrageenans also have great biomedical relevance in tissue development, since the use of GC in the preparation of hydroxyapatite-containing compounds (HAP) for tissue engineering purposes is well established [189,190]. Daniel-da-Silva et al. demonstrated that the composites prepared with *kappa*-CG fulfill some of the requirements of scaffold materials for bone regeneration, since they presented an interconnected porosity above 90% and pore diameter larger than 100 μm, essential for cell penetration and proper vascularization of the ingrown tissue [189].

Feng et al. reported that the incorporation of *kappa*-CG into a collagen-hydroxyapatite composite gel produces a composite material with the structural characteristics of natural bone, as the *kappa*-CG increased its compressive strength, and the composite material had good biocompatibility, making it a promising substitute material for the development of bone tissue [191]. In a recent study, Yegappan et al. developed an injectable CG nanocomposite hydrogel incorporated with whitlockite (the second most abundant bone mineral in humans) nanoparticles and dimethyloxallyl glycine (a proangiogenic drug) (Figure 10). The results showed that hydrogels are mechanically stable, cytocompatible, have better protein adsorption, and promote cell migration, whereas the proangiogenic drug acts as a chemo-attractant for endothelial cells. The proangiogenic drug produces an initial burst release followed by sustained release for 7 days. The authors therefore indicated that the incorporation of whitlockite and the proangiogenic drug in a CG hydrogel promoted osteogenesis and angiogenesis in vitro, which could potentially be harnessed in healing bone tissue engineering [128].

The combination of nanohydroxyapatite (nHA), gum arabic (GA), and *kappa*-CG in varying concentrations was recently evaluated for their application in tissue engineering. Comparative assessment of morphological, physicochemical, and biological studies revealed that ternary systems possess superior biocompatibility, protein adsorption and osteogenic protein expression as compared to the binary system formed by nHA and GA. Therefore, the authors concluded that the incorporation of *kappa*-CG to the ternary system outperforms the binary system collectively in terms of bioactivity with the maximal activity shown by the nanocomposite formed by nHA/GA/*kappa*-CG in a ratio of 60/20/20 [192].

CGs have also been considered in the development of an injectable bone substitute (IBS) [126,131]. Based on CG and nanohydroxyapatite (nHA), the formulations that consisted of 1%, 1.5%, and 2.5% CG and 60% nHA were evaluated, in order to determine its interactions with human osteoblasts. The results indicated a minimal degree of cytotoxicity, with a viability of more than 90% for all test samples, indicating that bone substitutes do not pose a toxic threat against human osteoblast cells, being effectively safe for bone cells [131].

Other research reports on the development of injectable bone substitutes based on CG mixed with different proportions of nanohydroxyapatite (nHA). The results show that CG is a potential vehicle for nHA particles, since it is able to aggregate them properly while maintaining the integrity of the injectable system and enabling good handling of the injectable bone substitutes with low and stable injectability forces; these are all promising characteristics for the administration of new injectable bone substitutes [193]. In other current research, the preparation of injectable bone substitutes using *kappa*-CG and nHA was described. The *kappa*-CG/nHA combination was not toxic, since cell viability was greater than 90%. The surfaces of this combination also proved to be adequate for permitting the adhesion of osteoblasts, since adhesion test results demonstrated an 88% or greater attachment of cells to IBS surfaces. Based on these results, the authors suggest that the combination of *kappa*-CG with nHA form suitable IBS, capable of inducing the proliferation and differentiation of cells with osteoblastic genotypes, demonstrating its potential for bone tissue engineering applications [126].

In a recent study, the development of rifampicin-loaded nanocomposites for tuberculosis osteomyelitis-infected tissues regeneration was analyzed. The nanocomposites consisted of *kappa*-CG, maleic anhydride, isoniazid, and nHA, loaded with rifampicin. The nanocomposites exhibited a spherical shape with 250 nm, being suitable for drug delivery that can easily flow through biological fluids without any interferences, with effective drug release occurring at pH 5.5, and the nanocomposites are non-cytotoxic while they induce cell activation and proliferation, being capable of inducing the growth of osteoblastic cells. In general, nanocomposites show remarkable characteristics for the regeneration of bones affected by tuberculosis [130].

CGs have been considered for growth factor delivery systems in a tissue engineering approach [194,195]. *Kappa*-CG was a suitable option for the development of growth factor carrier systems; the high encapsulation efficiency and the controlled administration profile allowed a sustained release of the incorporated bioactive factor [194]. 

In another study, a hydrogel based on two natural polysaccharides was prepared in aqueous medium with galactomannan and different concentrations of *kappa*-CG and CaCl_2_ for obtaining different pH values. The best formulation was obtained with 1.7% (*w*/*v*) galactomannan and 0.5% (*w*/*v*) *kappa*-CG, containing 0.2 M CaCl_2_ at pH 5.0. This hydrogel has good and stable physical properties. The SEM analysis of the hydrogel matrix showed an interconnected macropore architecture with a rough surface. These pore matrix interconnections may result in an easy flow for biomolecules, and produce scaffold matrices for tissue engineering [196].

Popa et al. evaluated the in vitro and in vivo biocompatibility of carrageenan-based hydrogels used in regenerative medicine and tissue engineering. The results indicated that *kappa*-CG hydrogels did not significantly affect L929 mouse fibroblast cell line metabolic activity. The contact of human polymorphonuclear neutrophil cells with *kappa*-CG resulted in a reduced and negligible signal in the detection of superoxide and hydroxyl anions, respectively. The in vivo experiments indicated that *kappa*-CG induces a low inflammatory response. Overall, *kappa*-CG hydrogels are biocompatible and can be useful in tissue engineering [197].

Soft tissue engineering has been developed as a new strategy for restoring diseased soft tissues and organs [198], but its application has certain limitations since it is usually accompanied by invasive surgery, which has some adverse effects, so injectable biomaterials are popular platforms to accurately fill the damaged area and facilitate tissue regeneration [199]. Mokhtari et al. developed a novel injectable nanohybrid hydrogel based on methacrylate-*kappa*-CG (KaMA)-dopamine functionalized graphene oxide (GOPD) for soft tissue engineering. This hydrogel was developed using a dual-crosslinking mechanism, which transmuted this hydrogel to unique biomaterials with superior properties. The incorporation of GOPD nanoparticles in the KaMA hydrogels allowed them to be efficiently injected by enhancing the proprieties of KaMA. The biocompatibility of this nanohybrid hydrogel was noticeably promoted by increasing the GOPD content; the authors thus concluded that the formulation could be a desirable choice for soft tissue engineering and 3-D bioprinting applications [200]. 

A novel synergistic *kappa*-CG/xanthan gum/gellan gum hydrogel film was developed. The results of the FT-IR, DSC, and SEM analysis showed a clear interaction between components to form a new material. The combination produces a synergistic increase in gel network strength and an improvement in gel firmness; this formulation had significantly improved thermal, mechanical, and water vapor permeability properties. The authors report that the film can be useful in various applications, such as in soft tissues, and particularly in topical skin applications, scaffolds, and tissue engineering [201]. 

There is a growing need to develop cartilaginous tissues. The chemical structure of CG makes it possible to mimic the sulphated glycosaminoglycans present in the extracellular matrix of cartilage, making these polymers potential candidates for cartilage development [22]. Popa E. et al. reported the potential use of hydrogels of *kappa*-CG for the supply of adipose tissue stem cells for cartilage regeneration. The results showed that hydrogels of *kappa*-CG can be an alternative cell delivery hydrogel system. Encapsulated human fat stem cells remained viable, and proliferated and differentiated in the chondrogenic lineage. The mechanical property of the hydrogel was also similar to that of native cartilage tissue. These promising in vitro results provide a basis for new studies into the treatment of cartilage defects [202]. 

#### 4.2.2. 3-D Bioprinting Applications 

The low availability of donor tissues and organs has led to the development of biofabrication of complex composite living tissues through three-dimensional bioprinting (3-D), an additive manufacturing process in which hydrogel biolinks loaded with live tissue cells are deposited layer by layer [203]. The fundamental components of 3-D bioprinting are cells, biomaterials, and biological factors. Biomaterials are key components since they must be compatible with cells, cell aggregates, microcarriers, etc., and must also have relevant mechanical and functional properties that allow the development of structures suitable for obtaining complex tissues [204]. Marine biomaterials for 3-D bioprinting applications have received widespread attention due to their biolinks in 3-D bioprinting. The most commonly used are alginate, CS [204], and carrageenan [22].

*Kappa*-CG was used as a biological binder for 3-D printing with CaP (calcium phosphate) paste [205]. 3-D bioprinting of a multilayer structure with a strong interfacial bond has also been tested by exploiting the electrostatic interaction between a cationic hydrogel (thixotropic gelatin) and an anionic hydrogel (*kappa*-CG). *Kappa*-CG and gelatin hydrogels were printed alternately using a 3-D bioprinter, which resulted in a 3-D construction called *kappa*-CG/gelatin. The results showed that the interfacial force between a gelatin hydrogel and a *kappa*-CG hydrogel was significantly greater than that of the bilayer gelatin hydrogel or the *kappa*-CG bilayer hydrogel. The bioprinted multilayer *kappa*-CG-gelatin hydrogel construction has very good biocompatibility and good structural integrity at 37 °C [206].

Kim et al. demonstrated the feasibility of the precise fabrication of alginate/carrageenan composite scaffolds using extrusion-based 3-D bioprinting. The rheological properties of alginate-based hydrogel improved with increased concentrations of carrageenan in the composite hydrogels, and was best with a carrageenan concentration of 1.5%. The results demonstrated excellent structural strength and printability of the carrageenan composite without any significant negative effects on cell viability, pointing to its application in the field of tissue engineering and regenerative medicine [207]. 

Three-dimensional bioprinting of oppositely charged hydrogels has also been reported. The study was based on three anionic hydrogels (alginate, xanthan, and *kappa*-CG) and three cationic hydrogels (chitosan, gelatin, and gelatin methacrylate (GelMA)) (Figure 11). The authors found that *kappa*-CG (2% *kappa*-CG hydrogel) and GelMA (10% GelMA hydrogel) are the best combination of oppositely charged hydrogels for 3-D printing. The interfacial bonding between a *kappa*-CG layer and a GelMA layer is significantly higher than that of bilayer *kappa*-CG or bilayer GelMA thanks to the formation of polyelectrolyte complexes. In addition, the bioprinted *kappa*-CG/GelMA construct demonstrated an excellent biocompatibility, with a cell viability of >96% [208].

In a recent study, 3-D scaffolds composed of mixtures of hydrogel, methacrylamide-modified gelatin, and methacrylate *kappa*-CG were developed using extrusion-based 3-D printing. The results indicated that scaffolds remain stable over time (21 days), are capable of absorbing large amounts of water, and exhibit mechanical properties comparable to those of native breast tissue. These findings imply that scaffolding can mimic the physicochemical characteristics of the natural extracellular matrix of native adipose tissue. They also obtained a similar cell viability (>90%) and a proliferation rate after 14 days for stem cells derived from adipose tissue when sown in both types of scaffolds [209]. 

## 5. Conclusions

In this review, we analyzed and described the various applications of carrageenans, in drug delivery, as bioactive agents and other significant biomedical applications, such as tissue engineering, including 3-D printing. The ability of carrageenans to form hydrogels has allowed their use to cover many fields of application, not only biomedical but also food. Studies have shown good bioactive properties capable of controlling viral infections, such as HPV, HSV, or SARS-CoV-2, bacterial infections, and even pathophysiological processes, such as hyperlipidemia, showing that CGs are highly safe, effective, and of course biocompatible, biodegradable, and non-toxic. These properties, in addition to the physicochemical properties, have led them to an increase their use and support their future applicability.

However, the different therapeutic applications are still in the experimental phase, there are certain limitations in the use of these sulfated polymers in the biomedical field, but the path is outlined and currently the development of new research focused on the use of carrageenans, such as the main components, is booming. It remains to be seen how these formulations behave in clinical trials, to verify their effectiveness and real potency.

## Figures and Tables

**Figure 1 marinedrugs-18-00583-f001:**
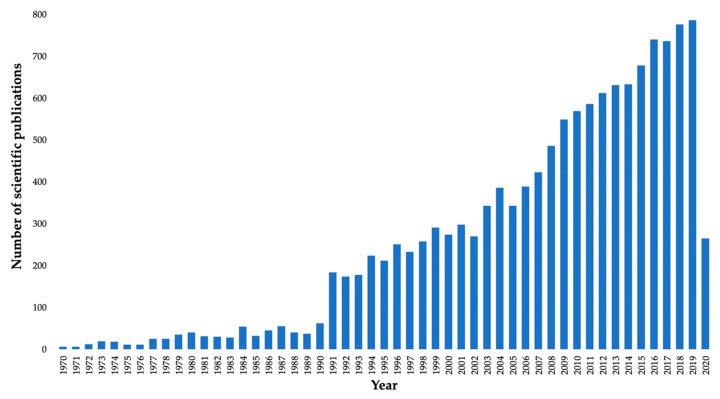
Number of scientific publications published on the topic “carrageenan” as a function of publication years. Taken from ISI Web of Knowledge, April 2020.

**Figure 2 marinedrugs-18-00583-f002:**
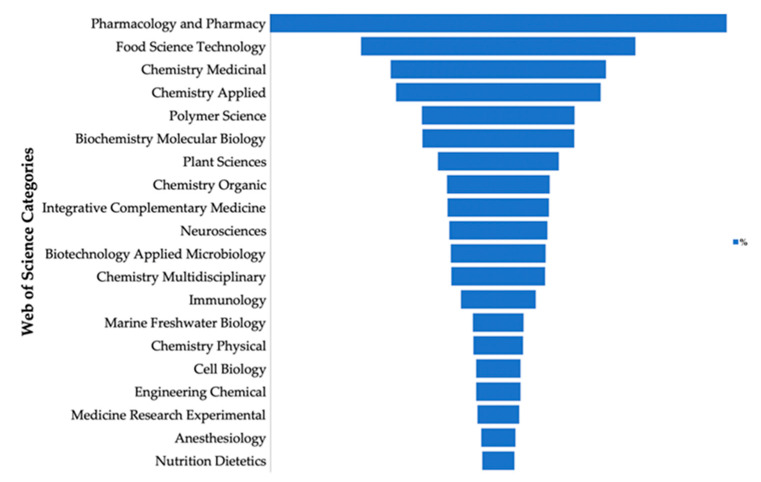
Percentage of categories according to the Web of Science. It should be noted that there are more categories than those shown in the figure, but they represent less than 2% of publications. Taken from ISI Web of Knowledge, April 2020.

**Figure 3 marinedrugs-18-00583-f003:**
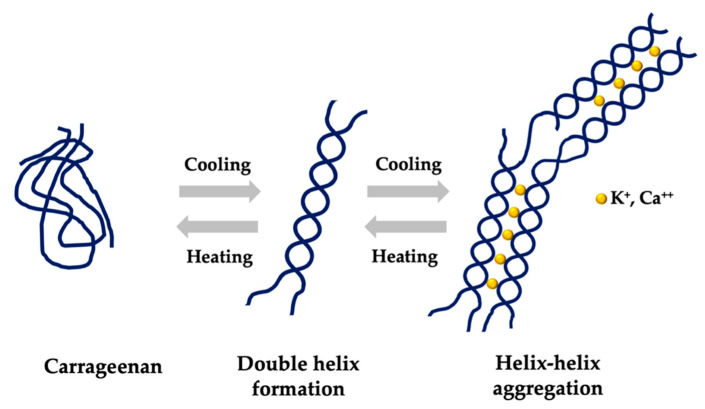
The possible gelling mechanism of *kappa*- and *iota*-CG. CG molecules form gels via two steps: the CG chains are in a coiled state in a hot solution, and they intertwine in double-helical structures as the solution cools. Finally, on further cooling, the double helices are thought to nest together with the aid of K^+^ or Ca^++^. Figure adapted from [40].

**Figure 4 marinedrugs-18-00583-f004:**
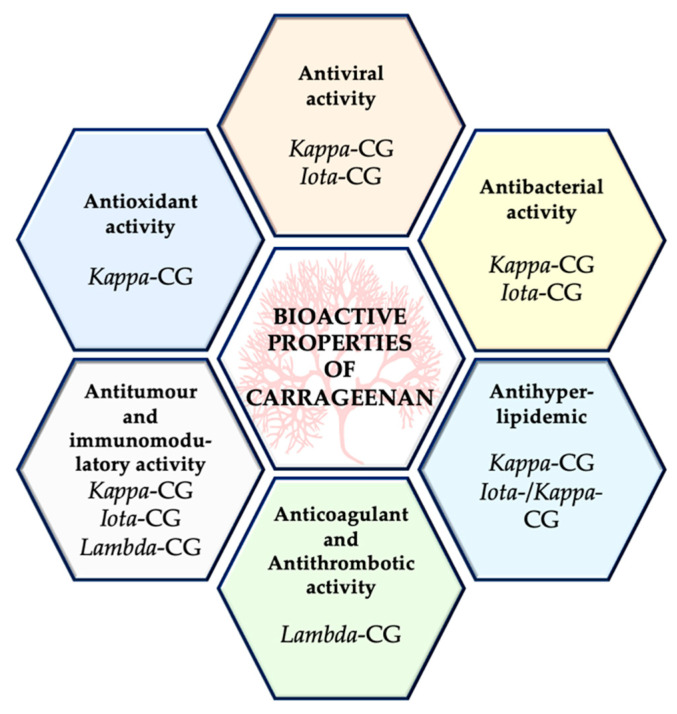
Bioactive properties of the different types of carrageenans used in biomedical applications [5,13,18].

**Figure 5 marinedrugs-18-00583-f005:**
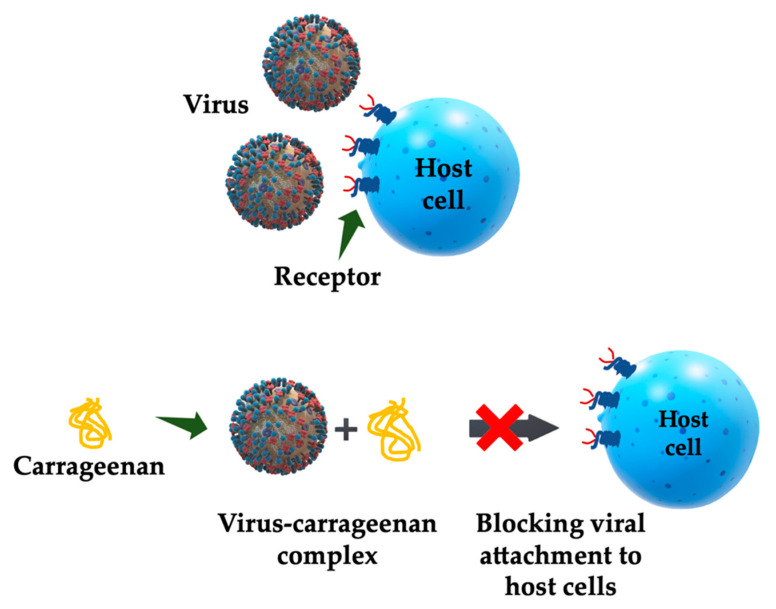
Mechanism of viral inhibition of carrageenans. Figure adapted from [59], with permission from Elsevier, Copyright © 2017.

**Figure 6 marinedrugs-18-00583-f006:**
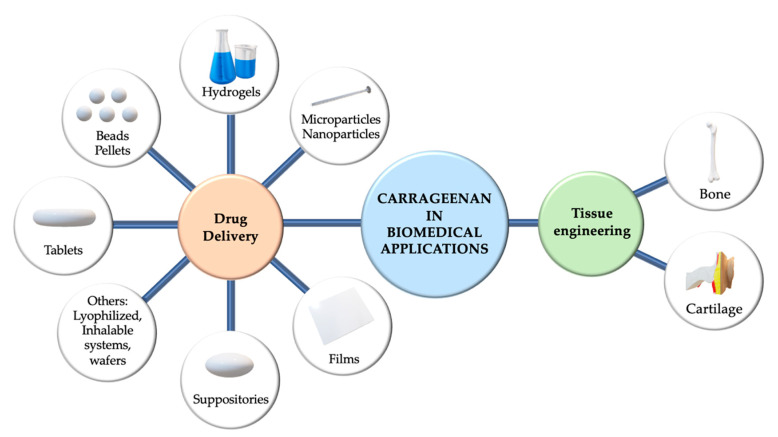
Carrageenan as a biomaterial for biomedical applications and in drug delivery systems and tissue engineering [22,106,108,112,121,126,132].

**Figure 7 marinedrugs-18-00583-f007:**
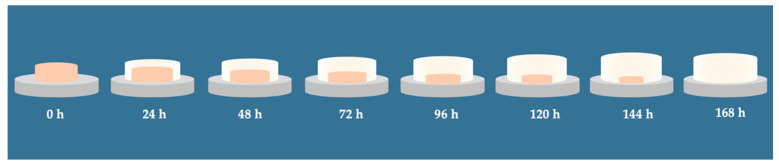
Swelling behavior of vaginal tablets based on *iota*-CG and HPMC over time. The combination of polymers produces an adequate uptake of the medium that allows them to develop the precise consistency and volume of gel for the controlled release of acyclovir [106].

**Figure 8 marinedrugs-18-00583-f008:**
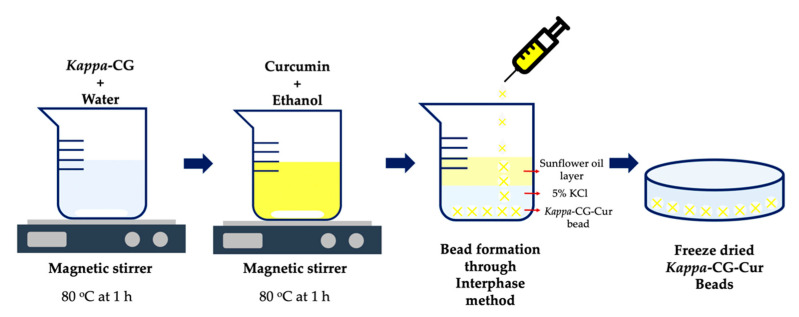
Preparation of curcumin-loaded *kappa*-CG beads using the interphase method. Figure adapted from [144], with permission from Elsevier, Copyright © 2017.

**Figure 9 marinedrugs-18-00583-f009:**
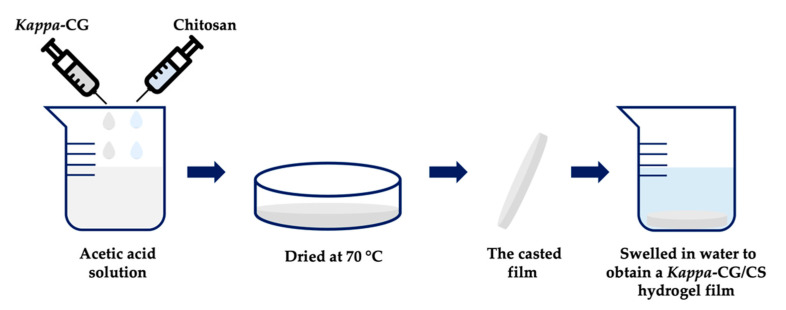
Diagram of the preparation of *kappa*-CG/CS hydrogel films. Figure adapted from [155], with permission from American Chemical Society, Copyright © 2018.

**Figure 10 marinedrugs-18-00583-f010:**
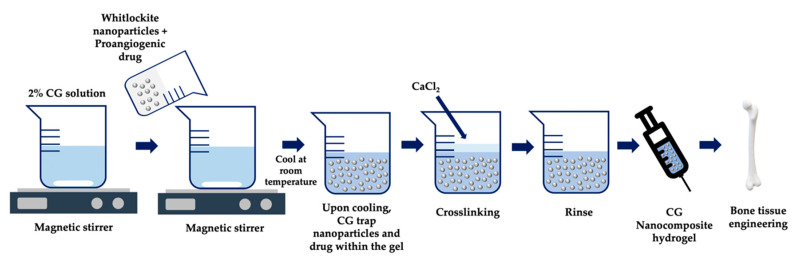
Preparation of nanocomposite hydrogel. Figure adapted from [128], with permission from Elsevier, Copyright © 2019.

**Figure 11 marinedrugs-18-00583-f011:**
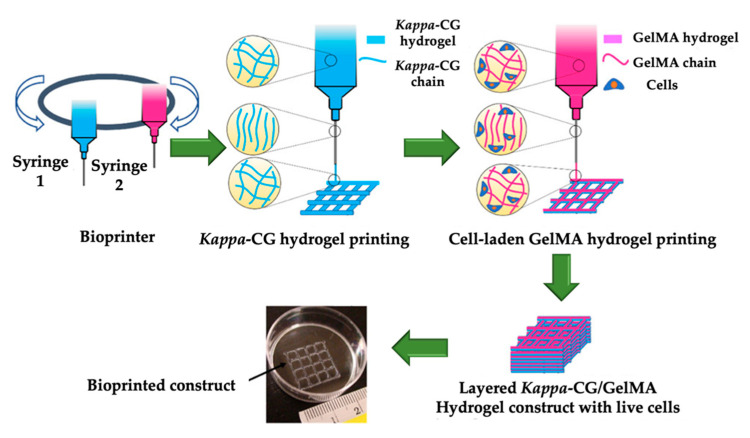
3-D bioprinting of *Kappa*-CG/GelMA hydrogel. Reproduced from [208], with permission from American Chemical Society, Copyright © 2018.

**Table 1 marinedrugs-18-00583-t001:** Characteristics of the main types of carrageenan [35].

Sulphated Polysaccharide	Marine Algae Group	Main Genera	Type of Carrageenan	Chemical Structure	Ester Sulphate Content (%)	3,6-AG Content (%)	Reference
Carrageenan	*Rhodophyceae*	*Chondrus*, *Euchema*, *Furcellaria*, *Fucus*, *Gigartina*, *Hypnea*, *Iridae*, *Kappaphycus*	*Kappa*-CG	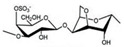	25–30	28–35	[35]
			*Iota*-CG	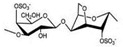	28–30	25–30	
			*Lambda*-CG	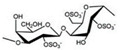	32–39	–	

**Table 2 marinedrugs-18-00583-t002:** Bioactive properties of CG and their biomedical applications.

Bioactive Properties	Type of CG	Applications	Reference
Antiviral activity	*Kappa*-CG *Iota*-CG	Inhibits Herpes Simplex Virus (HSV), Human Papillomavirus (HPV), Varicella Zoster Virus (VZV) and Human Rhinoviruses	[4,5,20,45,59,60]
	*Lambda*-CG/*kappa*-CG	Bioactivity against HPV and HSV-2	[64]
	CG	Genital HPV infection	[65]
	*Iota*-CG	Reduction in cold symptoms and reduces the growth of Human Rhinoviruses (HRV)	[66]
	*Iota*-CG	Potential inhibitor of the Influenza A Virus infection	[67]
	*Kappa*-CG	H1N1/2009 and other similar viruses	[68]
	*Iota*-CG *Kappa*-CG	Influenza A Virus strains (pandemic H1N1/09, H3N2, H5N1, H7N7)	[69]
	*Iota*-CG	Human Rhinovirus (HRV) 1a, hRV8 and Human Coronavirus OC43	[70]
	*Kappa*-CG	Enterovirus 71 (EV 71)	[71]
	*Lambda*-CG	Rabies Virus (RABV)	[72]
	*Kappa*-CG *Iota*-CG *Lambd**a*-CG	Varicella Zoster Virus (VZV)	[73]
	*Iota*-CG	Severe acute respiratory syndrome coronavirus 2 (SARS-CoV-2)	[74]
Antibacterial effects	*Iota*-CG	Inhibits the growth of the bacterial strains	[77]
	*Iota*-CG	Ocular *Chlamydia trachomatis* infection	[30]
	*Kappa*-CG	Activity against *Saccharomyces cerevisiae*	[78]
	*Kappa*-CG	Activity against Gram-positive and Gram-negative bacteria	[79]
	*Kappa*-CG	Reduced the production of interleukin-6 in cells treated with *kappa*-CG	[80]
	*Kappa*-CG	Activity against *S. aureus* and *E. coli*	[81]
	*Kappa*-CG	Activity against *S. aureus*, *Bacillus cereus*, *E. coli* and *Pseudomonas aeruginosa*	[82]
Antihyperlipidemic effects	CG	Hypocholesterolemic effect	[45]
	*Kappa*-CG	Reduces serum levels of total cholesterol, triglycerides and low-density lipoprotein cholesterol (LDL-C), and increasing high-density lipoprotein cholesterol (HDL-C)	[84,85]
	*Kappa*-CG *kappa*/β-CG *I**ota*/*kappa*-CG	Modulate prostaglandin E2 synthesis and stimulate IL-1β and IL-6 synthesis	[86]
	*Kappa*-CG/*iota*-CG	Reduces in serum levels of total cholesterol	[87]
	*CG*	Metabolic syndrome	[88]
	*Iota*-CG	Metabolic syndrome	[89]
Anticoagulant and antithrombotic activity	*Lambda*-CG	Highest anticoagulant activity in the rabbit whole blood test	[91]
	*Lambda*-CG	Antithrombotic activity	[4,21]
	*Kappa*-CG *Iota*-CG *Iota*/*nu*-CG *Theta*-CG *Lambda*-CG	Anticoagulant activity	[92]
Antitumor and immunomodulatory activity	*Lambda*-CG	Anticancer effects, immunomodulation	[95,97]
	*Lambda*-CG	Improve the antitumor activity of 5-Fluorouracil	[93]
	*Lambda*-CG	Inhibits tumour growth in mice with murine melanoma cell lines	[98]
	*Kappa*-CG *Lambda*-CG	*Kappa*-CG delays the cell cycle in the G2/M phase *Lambda*-CG stalled the cell cycle in both the G1 and G2/M phase	[99]
	*Iota*-CG	Suppressed tumour growth, induced apoptosis, and halted the G1 phase	[100]
	*Kappa*-CG *Iota*-CG	Cytotoxic effect on LM2 tumour cells	[101]
	*Lambda*-CG/*epsilon*-CG	Inhibits colorectal cancer stem-like cells	[102]
	*Kappa*-CG *Lambda*-CG	Antitumour and immunotropic effects	[103]
Antioxidant activity	*Kappa*-CG	Antioxidant activity in the multilayer coating	[105]

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
