# Peer review of "Carrageenan: Drug Delivery Systems and Other Biomedical Applications"

_marinedrugs, 2020, doi:10.3390/md18110583_

Round 1

Reviewer 1 Report

The paper is scientifically interesting, very current and with good bibliographic support. However, it has some imperfections that need correction.

line 58 and 59- Rhodophyceae family does not currently exist. I suggest replacing it with the Florideophyceae class. Consult the link:
https://www.algaebase.org/search/species/detail/?tc=accept&species_id=19519
I suggest ordering alphabetically; Agardhiella is repeated.
line 68 - like the FDA I suggest Generally Recognized As Safe (GRAS)
line 114 and table 1 confirm the references (39 or 40)?
line 145 - attention, if the figure is not original you must have authorization from the authors.
line 147 and 148 I suggest the replacement by Rhodophyta Phylum and consult the link above indicated.
line 381 - GAG means glycosaminoglycans, you must first spell it out.
line 452 - as previously I suggest placing (Figure 4) and confirming the number (Figure 5)??

Author Response

Response to Reviewer 1

line 58 and 59- Rhodophyceae family does not currently exist. I suggest replacing it with the Florideophyceae class. Consult the link:
https://www.algaebase.org/search/species/detail/?tc=accept&species_id=19519
I suggest ordering alphabetically; Agardhiella is repeated.

After consulting the suggested link, Rhodophyceae family has been replaced by Florideophyceae class. In addition, it was ordered alphabetically. (Line 52-53)

line 68 - like the FDA I suggest Generally Recognized As Safe (GRAS)

The sentence was changed and completed based on what the FDA mentions, referring to the Generally Recognized As Safe (GRAS) list. (Line 62-63)

line 114 and table 1 confirm the references (39 or 40)?

Fixed a syntax error, in the order of references previously listed as 39 and 40. The same ones that were renumbered to 33 y 34.

line 145 - attention, if the figure is not original you must have authorization from the authors.

Figure 3 was adapted from other previous articles regarding the possible gelling mechanism of carrageenans. The respective reference was placed. (Line 177)

line 147 and 148 I suggest the replacement by Rhodophyta Phylum and consult the link above indicated.

Rhodophyceae family was replaced by Phylum Rhodophyta. (Line 179)

line 381 - GAG means glycosaminoglycans, you must first spell it out.

In the sentence it was completed with the name glycosaminoglycans and placed in parentheses (GAG). (Line 422)

line 452 - as previously I suggest placing (Figure 4) and confirming the number (Figure 5)??

A syntax error was corrected when listing the figures within the text of the manuscript, currently remaining as figure 6. (Line 494)

Reviewer 2 Report

This manuscript is interesting, but its scientific relevance is not clear. It should be revised to improve it. A critical analysis from the authors concerning the studies reported in the literature should be made. More comments below:

  • Considering that most of the studies reported in the manuscript are mainly related to drug delivery system-based carrageenan matrices, the manuscript's title should be adapted to the content. Moreover, the manuscript should be re-organized.
  • The introduction should focus mainly on carrageenan and related relevant topics such as algae sources and its bioactivity. Therefore, the paragraphs between lines 27 -56 should be reformulated.
  • A table summarizing the recent studies involving carrageenan properties should be helpful for readers.
  • The addition of some carrageenan results as illustrations should be helpful, considering the high number of studies reported in the manuscript.
  • Page 11, line 440, please clarify the meaning of the paragraph.
  • Page 12, line 452, please check the reference to Figure 4.
  • Page 12, line 456-457, please, the idea is not complete.
  • Page 12, line 471-472, “ Theophyllline, sodium salicylate….. drugs. The matrices that contained CG were useful to produce controlled ….a zeo-order kinetic. Please add the reference to support the statement. Moreover, indicate which matrices were refereed? Please add the information.
  • Page 23, line 943-944, the paragraph is not relevant to the context, and it should be revised.
  • If the authors do not change the manuscript's focus, section 4 related to carrageenan -based tissue engineering should be reinforced.
  • The conclusion section should be revised. The future trends in carrageenan are not clear.
  • Please verify the reference format of the journal. There are many references without volume or final pages.

Author Response

Response to Reviewer 2

Considering that most of the studies reported in the manuscript are mainly related to drug delivery system-based carrageenan matrices, the manuscript's title should be adapted to the content. Moreover, the manuscript should be reorganized.

In view of the suggested, the title of the article is changed, taking into account that a specific description of the bioactive properties of carrageenans was made in the manuscript and how these acts directly as antiviral, antibacterial, antihyperlipidemic, anticoagulant, antithrombotic, antitumor, immunomodulator and antioxidant. Also describes different applications of carrageenan in the pharmaceutical field as drug delivery systems and in the biomedical field as materials used in tissue engineering, considering a focus on the pharmaceutical and biomedical field of all the work.

New title - Carrageenan: Drug delivery systems and other biomedical applications.

The introduction should focus mainly on carrageenan and related relevant topics such as algae sources and its bioactivity. Therefore, the paragraphs between lines 27 -56 should be reformulated.

Part of the introduction was modified, the sentence between lines 27 and 29 and the paragraphs between lines 45 and 56 were deleted, mentioning that within the context of the introduction it was sought to briefly mention the main characteristics of the most commercially used marine polymers ( alginate and agar), in order to mention a little the differences of carrageenans with those other polymers derived from algae and whose structure, although different, is used for applications similar to carrageenans.
In addition, it should be mentioned that within the manuscript there is a section that describes the sources of carrageenans and a detailed section of all the bioactive properties of carrageenans.

A table summarizing the recent studies involving carrageenan properties should be helpful for readers.

Based on the observation made, a table was developed that summarizes the bioactive properties of carrageenans and their applications in the biomedical field (Table 2). It is worth mentioning that there is a graphical scheme and Table 1 that describe certain properties of carrageenans, such as the origin and chemical structure.

The addition of some carrageenan results as illustrations should be helpful, considering the high number of studies reported in the manuscript.

The illustrations placed in the manuscript are my own and were developed from the referenced works. Two additional illustrations were added, the first illustration referring to the mechanism of antiviral action of carrageenans (Figure 5), and the second illustration referring to the use of hydrogels in 3D printing (Figure 11).

Page 11, line 440, please clarify the meaning of the paragraph.

The indicated paragraph was completely modified, based on the suggestion. This paragraph explains the transformation of excipients into multifunctional components that not only contribute to the drug manufacturing process, but also certain polymeric components such as carrageenans provide bioactive and functional properties to pharmaceutical forms. (Line 481)

Page 12, line 452, please check the reference to Figure 4.

A syntax error was corrected at the time of listing the figures within the manuscript, being listed as figure 6. (Line 494)

Page 12, line 456-457, please, the idea is not complete.

The sentence corresponding to lines 456-457 was deleted, and a new sentence was developed that indicates the relationship of carrageenans to tissue engineering. (Line 498-499)

Page 12, line 471-472, “Theophyllline, sodium salicylate….. drugs. The matrices that contained CG were useful to produce controlled ….a zeo-order kinetic. Please add the reference to support the statement. Moreover, indicate which matrices were refereed? Please add the information.

The information requested based on the cited article was added. (Line 528-531)

Page 23, line 943-944, the paragraph is not relevant to the context, and it should be revised.

The sentence corresponding to lines 943-944 was deleted, agreeing that it is not relevant to the context.

If the authors do not change the manuscript's focus, section 4 related to carrageenan -based tissue engineering should be reinforced.

References and description of the topic were increased. (Line 1013-1070).

The conclusion section should be revised. The future trends in carrageenan are not clear.

The conclusion was revised and restructured.

Please verify the reference format of the journal. There are many references without volume or final pages.

The references were reviewed, and the missing information was completed.

Round 2

Reviewer 2 Report

 The revised version of the manuscript is better than the previous one; however, it still needs improvements to improve its relevance scientific. 

  • Please revise the introduction section to focus on relevant topics such as algae sources and its bioactivity. Therefore, the paragraphs between lines 37 and 43 should be eliminated.
  • A critical analysis from the authors concerning the studies reported in the literature is still missing in several parts of the manuscript, mainly in section 3.
  • The conclusion section should be further revised. The future trends that do not give the role of carrageenan will have in drug development and biomedical purposes.

Figure 5, please replace the term “graph” in the legend for Figure

Author Response

Response to Reviewer 2 (Round 2)

Please review the introduction section to focus on relevant topics such as algae sources and its bioactivity. Therefore, the paragraphs between lines 37 and 43 should be eliminated.

Part of the introduction was modified, the sentence between lines 37 and 43 was eliminated. New information related to the sources of the carrageenans was added and more bibliographic references on this topic were added. (Lines 34-40).

A critical analysis from the authors concerning the studies reported in the literature is still missing in several parts of the manuscript, mainly in section 3.

In view of your suggestions, a critical analysis of section 3 was carried out.

The conclusion section should be further revised. The future trends that do not give the role of carrageenan will have in drug development and biomedical purposes.

The conclusion was completely changed. (Lines 1334-1347).

Figure 5, please replace the term "graph" in the legend for Figure

The graphic term was changed in the indicated place.

Round 3

Reviewer 2 Report

The revised version of the manuscript is better than the previous one. However, minor revisions should be made before accept for publication.

page 27, line 1036, The results indicate that kappa-CG/nHA bone substitutes are a biomaterial with a great capacity ..please correct the paragraph.

Conclusions section

page 29 line 1147 " However, the different therapeutic applications are still in the clinical phase".. According to the studies' description, there are no many references for studies in the clinical phase. Please change it accordingly.

Author Response

The revised version of the manuscript is better than the previous one. However, minor revisions should be made before accept for publication.

page 27, line 1036, The results indicate that kappa-CG/nHA bone substitutes are a biomaterial with a great capacity ..please correct the paragraph.

The paragraph between lines 1034 and 1038 on page 27 was deleted and the entire paragraph was restructured.

Conclusions section

page 29 line 1147 " However, the different therapeutic applications are still in the clinical phase".. According to the studies' description, there are no many references for studies in the clinical phase. Please change it accordingly.

The errors indicated in the conclusions section were corrected.
